# Conformation transitions of the polypeptide-binding pocket support an active substrate release from Hsp70s

Jiao Yang[1], Yinong Zong[1], Jiayue Su[1], Hongtao Li[1], Huanyu Zhu[1], Linda Columbus[2], Lei Zhou[1] & Qinglian Liu[1]

Cellular protein homeostasis depends on heat shock proteins 70 kDa (Hsp70s), a class of ubiquitous and highly conserved molecular chaperone. Key to the chaperone activity is an ATP-induced allosteric regulation of polypeptide substrate binding and release. To illuminate the molecular mechanism of this allosteric coupling, here we present a novel crystal structure of an intact human BiP, an essential Hsp70 in ER, in an ATP-bound state. Strikingly, the polypeptide-binding pocket is completely closed, seemingly excluding any substrate binding. Our FRET, biochemical and EPR analysis suggests that this fully closed conformation is the major conformation for the ATP-bound state in solution, providing evidence for an active release of bound polypeptide substrates following ATP binding. The Hsp40 co-chaperone converts this fully closed conformation to an open conformation to initiate productive substrate binding. Taken together, this study provided a mechanistic understanding of the dynamic nature of the polypeptide-binding pocket in the Hsp70 chaperone cycle.

[1] Department of Physiology and Biophysics, Virginia Commonwealth University, Richmond, VA 23298, USA. [2] Department of Chemistry, University of Virginia, Charlottesville, VA 22908, USA. Correspondence and requests for materials should be addressed to Q.L. (email: qinglian.liu@vcuhealth.org)

As the most abundant molecular chaperones across all kingdoms of life, Hsp70s are the key players in every aspect of cellular protein homeostasis, including protein folding, assembly, transportation across membranes, degradation, and quality control[1–5]. With more than 40% sequence identity between Hsp70s in *E. coli* and human, Hsp70s are highly conserved, indicating an overall conserved mechanism[6]. All Hsp70s have two functional domains: a nucleotide-binding domain (NBD) at the N-terminus and a substrate-binding domain (SBD) at the C-terminus. Through binding and hydrolyzing ATP, the NBD provides energy to power efficient chaperone activity. The SBD is the site where polypeptide substrates bind in extended conformation in all the processes of protein homeostasis. In addition, the chaperone activity of Hsp70s is facilitated by two classes of co-chaperones: Hsp40s and nucleotide-exchange factors (NEFs)[1, 2, 7–9]. It is well established that NEFs speed up the release of ADP and the rebinding of ATP to Hsp70s after ATP hydrolysis. Essential and functionally conserved, Hsp40s (also called J-proteins) have been shown to stimulate the ATP hydrolysis step of Hsp70s. Some Hsp40s can directly bind both unfolded and partially folded polypeptides and have thus been proposed to bring polypeptide substrates to Hsp70s.

Extensive structural efforts for the past three decades have yielded a number of isolated domain structures from both prokaryotic and eukaryotic Hsp70s[10–17], demonstrating the conserved structural basis of each domain in binding its substrates. The NBD is composed of two big lobes, with the nucleotide-binding site in between. The SBD contains two subdomains: SBDβ and SBDα. All available SBD structures show that a single strand of peptide substrate in extended conformation binds to a polypeptide-binding pocket formed between two loops in SBDβ, $L_{1,2}$, and $L_{3,4}$[14–17].

As there is little contact between the two domains in the ADP-bound and nucleotide-free (apo) states, the isolated SBD structures represent these states and exhibit high affinity and slow kinetics for peptide substrates[18–21]. In contrast, ATP binding allosterically speeds up the kinetics of peptide substrate binding and release, and decreases the affinity by 2–3 orders of magnitude due to a more accelerated release rate[22, 23]. Various biochemical, NMR, and FRET studies have shown that ATP binding induces extensive interactions between the NBD and SBD, resulting in drastic conformational changes in the SBD that accelerate peptide substrate binding and release[1, 18, 19, 24–26]. The chaperone activity strictly depends on this ATP-induced allosteric coupling. The molecular mechanism of this essential allosteric coupling has thus been highly sought.

Following the initial breakthrough in crystallizing a full-length Hsp70 homolog in complex with ATP[27], three recent crystal structures of intact Hsp70s in the ATP-bound state and NMR studies have provided revolutionary insights in this essential mechanism[28–31]. In conjunction with a number of solution studies[19, 24, 32, 33], these structural studies support an overall uniform conformation for the ATP-bound state with relatively stable contacts between the two domains. Consistent with these structural studies, a structure of Ssb, a specialized Hsp70 in fungi, in complex with ATP has been published recently[34]. Although they disagree in details, the three recent crystal structures of classic Hsp70s in the ATP-bound state, including two structures of DnaK, an *E. coli* Hsp70, and one structure of human BiP, an Hsp70 in the endoplasmic reticulum (ER), have shown an open conformation of the polypeptide-binding pocket, which provides an explanation for the reduced affinity and fast kinetics[28, 29, 31]. But, is this the only conformation of the ATP-bound state for Hsp70s? It is well established that proteins are dynamic in solution. More importantly, it has been difficult to explain a productive chaperone cycle with only this conformation for the ATP-

bound state. Both binding and release of polypeptide substrates were proposed to occur in the ATP-bound state based on fast kinetics[1, 2, 22, 23]. How does Hsp70-ATP decide when to bind and when to release substrates with this single open conformation of the polypeptide-binding pocket?

Previous biochemical and NMR studies have shown increased conformational dynamics in the SBDβ region of the ATP-bound state[18, 19, 25, 30, 35, 36], indicating the possibility of other conformations. Moreover, all published Hsp70-ATP structures rely on the ATPase-deficient T199A mutation in the NBD of DnaK (T229A in BiP) alongside either an engineered disulfide bond or a loop mutation in SBD[28, 29, 31, 34]. This T199A mutation not only knocks out chaperone activity[37] but also completely abolishes Hsp40 co-chaperone interaction[38], suggesting that this mutation may lock Hsp70 in one of the conformations for the ATP-bound state.

To approach this key question, we have crystallized a full-length human BiP, a classic and essential Hsp70 in the ER[39–42], without the T229A mutation in the presence of ATP. This structure unexpectedly reveals a completely closed conformation of the polypeptide-binding pocket, which appears to be the dominant conformation for the ATP-bound state in solution. We also find that the binding of the Hsp40 co-chaperone efficiently converts this fully closed conformation to an open conformation. We propose that ATP binding induces the active release of bound polypeptide substrate to prevent nonproductive chaperone cycle, and that the Hsp40 co-chaperone is crucial to initiate efficient substrate binding to start a new productive chaperone cycle.

## Results

**A novel crystal structure of BiP in complex with ATP.** We have obtained a new crystal structure of human BiP containing both functional domains at 1.85 Å in the presence of ATP (Fig. 1, Table 1, and Supplementary Figs. 1, 2a–c). Unlike all previously reported Hsp70-ATP structures[28, 29, 31, 34], this new BiP-ATP structure, BiP-ATP2, has an entirely wild-type (WT) NBD. It

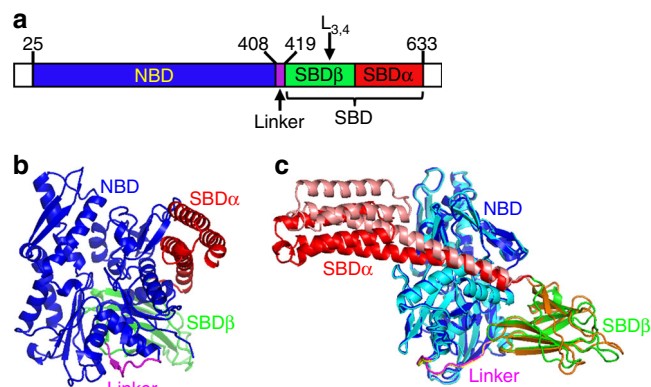

**Fig. 1** A new crystal structure of BiP in complex with ATP, BiP-ATP2. **a** Schematic of human BiP domain structure. The positions for key elements are indicated. The first 24 residues (signal sequence) and the last 20 residues (largely disordered) are not colored. **b** Ribbon diagram of the BiP-ATP2 structure. Domain color assignments are: NBD (blue), SBDβ (green), SBDα (red), and inter-domain linker (purple). **c** Overall structural comparison of BiP-ATP2 with BiP-ATP. BiP-ATP2 is colored as in **b**. The domain coloring for BiP-ATP (PDB code: 5E84) is: NBD (cyan), SBDβ (orange), SBDα (pink), and inter-domain linker (yellow). All the structural representations were prepared with PyMOL (The PyMOL Molecular Graphics System)

**Table 1 Data collection, phasing, and refinement statistics**

|  | Native | Se-MAD |
|---|---|---|
| Data collection |  |  |
| Space group | P1 | P1 |
| Cell dimensions |  |  |
| $a, b, c$ (Å) | 75.830, 75.752, 78.792 | 75.689, 75.702, 78.584 |
| $\alpha, \beta, \gamma$ (°) | 62.046, 62.235, 73.416 | 62.344, 62.243, 73.766 |
| Wavelength | 0.979 | 0.979 |
| Resolution (Å) | 50–1.85 (1.88–1.85) | 50–2.0 (2.03–2.00) |
| $R_{sym}$ or $R_{merge}$ | 0.070 (0.249) | 0.033 (0.152) |
| $I/\sigma I$ | 29.3 (3.3) | 21.2 (3.2) |
| Completeness (%) | 97.8 (96.5) | 97.6 (94.0) |
| Redundancy | 5.7 (3.3) | 1.9 (1.7) |
| cc1/2 | (0.916) |  |
| Refinement |  |  |
| Resolution (Å) | 39.09–1.85 |  |
| No. of reflections | 113887 |  |
| $R_{work}/R_{free}$ | 15.8%/18.9% |  |
| No. of atoms | 10,997 |  |
| Protein | 9588 |  |
| ATP | 62 |  |
| Water | 1209 |  |
| $B$-factors | 32.1 |  |
| Protein | 29.9 |  |
| ATP | 26.8 |  |
| Water | 37.9 |  |
| R.m.s. deviations |  |  |
| Bond lengths (Å) | 0.009 |  |
| Bond angles (°) | 1.202 |  |

*Note*: One crystal was used for each structure or data set; values in parentheses are for the highest-resolution shell

To obtain accurate phase information, single-wavelength anomalous diffraction (SAD) with selenomethionine (SeMet) protein was used to solve the structure (Table 1). The final model was refined at 1.85 Å against a native data set with good stereochemistry. The resulting electron density maps had high quality (Supplementary Fig. 2a, b).

The most remarkable difference between BiP-ATP2 and all previous Hsp70-ATP structures is the conformation of the polypeptide-binding pocket (a detailed comparison is shown below); in contrast, the overall conformation is highly similar (Fig. 1b–c, Supplementary Fig. 2c, d). For comparison we mainly used the BiP-ATP structure[28], which is almost identical to the two available DnaK-ATP structures[29, 31]. As pointed above, NBD shares the highest conformational identity (Fig. 1b–c, Supplementary Fig. 2c–e), confirming the high conservation in the ATP-binding site. At the same time, the relative orientation between NBD and SBDβ and their interface are almost identical, consistent with the uniform overall conformation of the ATP-bound state suggested by various NMR and FRET analysis[19, 24, 30, 32, 33]. The N-terminal segment of Helix A/B in SBDα is involved in contact with NBD, and this interface is also almost identical to BiP-ATP. However, the position for the rest of SBDα is shifted noticeably, which could be due to crystal packing (Fig. 1c, Supplementary Fig. 2d, g, h). Interestingly, BiP in the BiP-ATP2 structure packs as a dimer in the asymmetric unit similar to those of DnaK-ATP and BiP-ATP[28, 29, 44], but the interface between the NBDs has shifted (Supplementary Fig. 2g, h). In the DnaK-ATP and BiP-ATP structures, the SBDα from one molecule docks on one side of NBD from the other molecule and stabilizes the formation of dimer. In contrast, this interaction is lost in BiP-ATP2, which could influence the position of SBDα.

does not carry the ATPase-deficient T229A mutation (T199A in DnaK)[28, 37], which has helped maintain the ATP-bound state in all previously reported Hsp70-ATP structures[28, 29, 31, 34]. Even with an intact NBD in the BiP-ATP2 structure, ATP is bound in the nucleotide-binding pocket (Supplementary Fig. 2b) and the NBD assumes an almost identical conformation as those seen in all three previously reported Hsp70-ATP structures (Fig. 1b, c, Supplementary Fig. 2d, e), supporting the notion that the T229A mutation has little impact on the overall conformation of NBD. One reason that may account for the lack of significant ATP hydrolysis with a WT NBD in this construct is the crystallization condition which contains 100 mM phosphate. As a product of ATP hydrolysis, phosphate at this concentration reduces the ATPase activity of BiP by about five folds (Supplementary Fig. 2f).

Besides the N-terminal signal sequence and C-terminal unstructured region, this new BiP structure carries only one modification from the WT form, the $L_{3,4}'$ modification ($L_{3,4}$ was replaced with a shortened sequence: TASDNQP → VGG) (Fig. 1a, Supplementary Fig. 1). This modification was characterized previously as maintaining purified BiP protein in monomeric form for crystallization[28, 29]. As shown previously[28], although this $L_{3,4}'$ modification significantly reduces BiP's affinity for a well-characterized model peptide substrate NR (sequence NRLLLTG[14, 43]), it has no appreciable impact on the structural integrity of the isolated SBD (Supplementary Fig. 3). Moreover, in the isolated SBD structure carrying this modification, the NR peptide binds to the polypeptide-binding pocket in a virtually identical way as that of the WT BiP, suggesting that the $L_{3,4}'$ modification has no appreciable effect on the polypeptide-binding pocket besides the shortened $L_{3,4}$ sequence.

**Conformation of the polypeptide-binding pocket**. The most striking feature of this new BiP-ATP2 structure is the conformation of $L_{1,2}$, one of the two peptide-binding loops (Fig. 2a). In all previously reported structures, regardless of nucleotide-binding states or constructs, $L_{1,2}$ assumes an almost identical conformation, projecting outward to allow peptide substrate binding (as shown for BiP-ATP in Fig. 2b)[14–17, 20, 28, 29, 31, 34, 45, 46]. This projecting-outward conformation of $L_{1,2}$ is supported by its buttressing loop $L_{4,5}$. In sharp contrast, $L_{1,2}$ in BiP-ATP2 surprisingly closed onto the polypeptide-binding site, packing against β3 and β4, which form the bottom of the polypeptide-binding pocket (Fig. 2a, d). The adjacent β1 and β2 also shifted with $L_{1,2}$. At the same time, $L_{1,2}$ is detached from $L_{4,5}$ while the conformation of $L_{4,5}$ is largely unchanged. As a result, the polypeptide-binding pocket almost completely disappears in BiP-ATP2 in contrast to the wide-open conformation of the polypeptide-binding pocket in BiP-ATP and DnaK-ATP structures.

In the previously reported BiP-ATP structure[28], the $L_{3,4}$ side of the polypeptide-binding pocket including $L_{3,4}$, $L_{5,6}$, and $L_{7,8}$ flipped open relative to the isolated SBD structure (Fig. 2b). Moreover, the $L_{3,4}$ side of both β3 and β4 strands is shifted downward to further open the polypeptide-binding pocket. These changes were maintained in the new BiP-ATP2 structure (Fig. 2a–c), consistent with the hypothesis that these conserved changes are caused by the NBD-SBDβ interaction. Taken together, relative to the BiP-ATP structure, $L_{1,2}$ in BiP-ATP2 rotated more than 60° (Fig. 2a–c). A number of residues in the β1-$L_{1,2}$-β2 segment have shifted significantly with greater than 10 Å shift at the Cα atoms of Gly431 (Fig. 2a–c, Supplementary Fig. 4c). Similar changes were observed relative to the DnaK-ATP structure (Supplementary Fig. 4a–c).

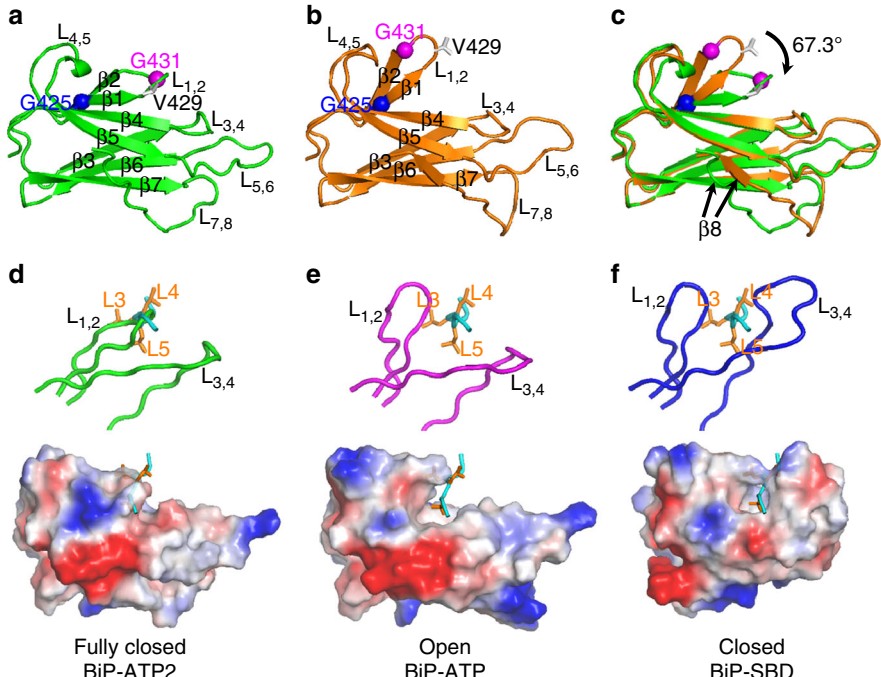

**Fig. 2** A novel conformation of the polypeptide-binding pocket revealed by BiP-ATP2. **a–c** Comparison of SBDβ conformations in BiP-ATP2 and BiP-ATP. Ribbon diagrams are drawn for SBDβs from BiP-ATP2 (**a**, green), BiP-ATP (**b**, orange; PDB code: 5E84), and their superposition (**c**). SBDβs were superimposed based on Cαs of β3–β7. Cα atoms of Gly425 and Gly431 are highlighted as blue and purple balls, respectively. The side-chains of Val429 are shown in stick representation. **d–f** Comparison of the polypeptide-binding sites in BiP-ATP2 (**d**), BiP-ATP (**e**), and isolated SBD of BiP (**f**, PDB code: 5E85). The top panels show the peptide-binding loops, $L_{1,2}$ and $L_{3,4}$ (in backbone worm representation). The surface of SBDβs are shown in the bottom panels (negative charge: blue; positive charge: red; and hydrophobic: white. The surfaces with qualitative electrostatic representation were generated by PyMOL using the "generate vacuum electrostatics" function.). The NR peptide bound in the isolated SBD is shown in cyan with the side chains of Leu3, Leu4, and Leu5 (orange) highlighted as stick drawings. BiP-ATP2 and BiP-ATP were superimposed as in **a–c**; BiP SBD was superimposed to BiP-ATP based on Cαs of β1-β2. The NR peptides in **d–e** were from **f**, and the positions were based on superposition. L loop, β β strand

In the isolated SBD structure, both $L_{1,2}$ and $L_{3,4}$ project outward to form the polypeptide-binding pocket that cradles the bound substrate (closed conformation; Fig. 2f)[14, 15, 28]. With only $L_{1,2}$ projecting outward and $L_{3,4}$ flipping out and open, the polypeptide-binding pocket in BiP-ATP is wide open to bind peptide substrates (open conformation; Fig. 2e), consistent with the fast kinetics and low affinity for the ATP-bound state. In contrast, when closed onto β3/β4, $L_{1,2}$ in BiP-ATP2 occupies the space for peptide substrates, and the polypeptide-binding pocket is almost completely buried (fully closed conformation; Fig. 2d). The residues lining the polypeptide-binding pocket become packed against each other and appear to exclude any substrate binding (Supplementary Fig. 4d–f).

Sse1 is an Hsp110 in yeast, a distant homolog of Hsp70s[47]. Interestingly, the fully closed conformation of the polypeptide-binding pocket described here is highly similar to our previously published Sse1-ATP structure (Supplementary Fig. 4g–i)[27], supporting structural conservation between Hsp70s and Hsp110s. Moreover, consistent with the fully closed conformation of the polypeptide-binding pocket, Sse1 does not show any significant binding to classic peptide substrates for Hsp70s, such as the NR peptide[48].

Crystal contacts are indispensable for crystal formation. In the BiP-ATP2 structure, only one weak hydrophobic contact was formed between $L_{1,2}$ and the symmetry mates, although a number of crystal contacts were formed around the SBDβ (Supplementary Fig. 5a–d). In contrast, more extensive crystal contacts were formed around $L_{1,2}$ in the two DnaK-ATP structures (Supplementary Fig. 5e–h). Although crystal contacts involving $L_{1,2}$ in these two DnaK-ATP structures are different, $L_{1,2}$ adopts a

practically identical conformation, suggesting that crystal contacts have no appreciable influence on the conformation of $L_{1,2}$. Except for this new BiP-ATP2 structure, all the ATP-bound structures of Hsp70s, including both DnaK and BiP, depend on the ATPase-deficient T199A mutation in DnaK (T229A in BiP)[28, 29, 31, 34], which could be the determining factor for the conformation of $L_{1,2}$ in these structures.

Another interesting feature of the SBDβ is the conformation of β8. Compared to BiP-ATP and DnaK-ATP, not only is the position of β8 shifted in BiP-ATP2 (Fig. 2c, Supplementary Fig. 4b), but the side-chain orientations are also changed (Supplementary Fig. 6a–d). BiP-ATP and DnaK-ATP have a similar β8 position (Supplementary Fig. 6f), and the β8 position and side-chain orientation in the isolated structures of DnaK and BiP are almost identical (Supplementary Fig. 6c, e). The β8 in BiP-ATP2 has similar side-chain orientations as those in the isolated BiP-SBD, whereas the side-chain orientations of β8 in BiP-ATP are flipped (Supplementary Fig. 6a–c). Interestingly, the first two residues at the N-terminal end of β8 in DnaK-ATP have similar side-chain orientations as the isolated SBD structures, but for the rest of β8 starting from the third residue, the side-chain orientations are flipped as those in BiP-ATP (Supplementary Fig. 6d, e). Thus, the side-chain orientations of β8 in DnaK-ATP seem to be a hybrid between BiP-ATP and BiP-ATP2. These differences in the ATP-bound structures suggest high conformational dynamics of β8 in response to ATP binding, and support the pivotal role of β8 in allosteric coupling[35, 49]. Moreover, the side-chain orientation changes may provide an explanation for the effect of mild mutations at Ile501 of DnaK in SBDβ dynamics and allosteric coupling[35].

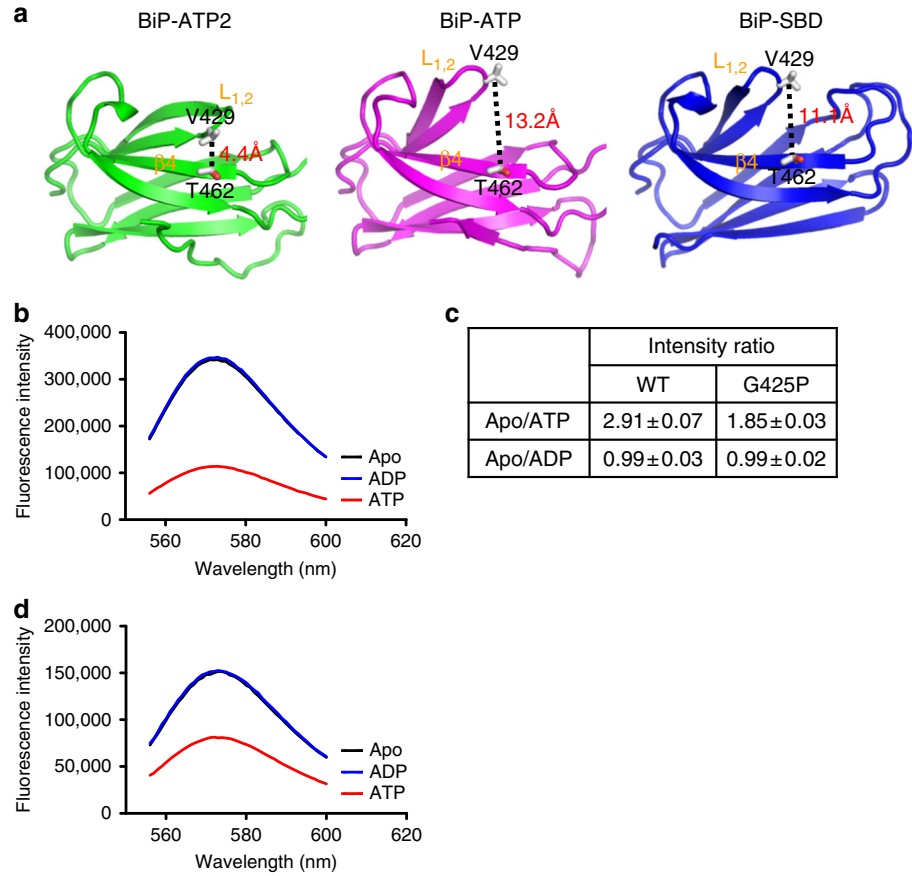

**Fig. 3** The fully closed conformation of $L_{1,2}$ exists as the major form in solution for BiP in the ATP-bound state. **a** The distances between Val429 (on $L_{1,2}$) and Thr462 (on β4) in the BiP structures. Only SBDβs are shown for BiP-ATP2 (left panel), BiP-ATP (middle panel), and isolated SBD (right panel). The structures were superimposed as in Fig. 2d–f. Val429 and Thr462 are highlighted as stick drawings. The distance between the Cβ atoms of Val429 and Thr462 are labeled. **b** ATP binding significantly quenched the fluorescence of the TMR-labeled BiP protein. Emission spectra were collected on the TMR-labeled BiP protein after addition of ATP (red), ADP (blue), or no nucleotide (Apo, black) with excitation at 547 nm. **c** Fluorescence changes in response to nucleotides for the TMR-labeled BiP proteins. Emission spectra were collected as in **b**, and the peak value at 573 nm was used for each emission spectrum. Each value was an average of five independent measurements from more than two preparations of labeled proteins. WT: the TMR-labeled BiP protein; G425P: the TMR-labeled BiP protein carrying the G425P mutation. **d** The ATP-induced quenching of the TMR-labeled BiP protein was significantly compromised by the G425P mutation. The assay was carried the same way as in **b** on the TMR-labeled BiP FRET protein carrying the G425P mutation

**The fully closed conformation of $L_{1,2}$ is the dominant form**. After observing this conformation of $L_{1,2}$ in the BiP-ATP2 structure, we tested whether this conformation exists in solution using FRET. To label BiP protein for FRET analysis, we have changed two residues to cysteine: Val429 in $L_{1,2}$ and Thr462 in β4 (Fig. 3a). BiP has two endogenous cysteine residues (Cys41 and Cys420). To prevent background labeling, we have mutated both to alanine together. The resulting BiP FRET construct BiP-C41A/C420A/V429C/T462C was purified to high quality. To characterize the influence of these modifications on BiP protein, we have analyzed the three key biochemical activities of BiP: ATPase, peptide substrate binding, and allosteric coupling. We used the well-established single-turnover ATPase and fluorescence polarization assays to determine the ATPase and peptide substrate binding activities, respectively[28]. For allosteric coupling, we have used the stimulation of ATPase activity by the model peptide NR. As shown in Supplementary Fig. 7, all these biochemical activities were largely intact for this BiP FRET construct, suggesting that none of the modifications has an appreciable effect on the biochemical activities of BiP.

In the BiP-ATP2 structure, due to the packing of $L_{1,2}$ against β4, the distance between V429 and T462 is quite close (4.4 Å for cβ) (Fig. 3a). In contrast, in the previously reported BiP-ATP

structure, this distance is much larger (13.2 Å) due to the projecting outward conformation of $L_{1,2}$. With the almost identical projecting outward conformation of $L_{1,2}$, a similar large distance (11.1 Å) was observed for the isolated SBD structure of BiP, which represents the ADP-bound and apo conformation. We took advantage of this difference in distance and labeled the BiP FRET construct with tetramethylrhodamine (TMR) following the protocol by Dr. Webb[50]. When the two molecules of TMR label are in close proximity such as in the BiP-ATP2 structure, they tend to stack on each other resulting in significant quenching in fluorescence intensity; whereas when the two labels are far enough to separate apart from stacking such as in the BiP-ATP structure or the isolated SBD structure, they behave independently, and fluorescence should be strong with little quenching.

If the fully closed conformation of $L_{1,2}$ observed in the BiP-ATP2 structure exists in solution, we will see quenching of fluorescence intensity upon addition of ATP. If this conformation is the dominant conformation for the ATP-bound state, we will see significant reduction of fluorescence intensity upon ATP binding. The BiP FRET protein was purified under reducing conditions. After labeling with TMR, complete labeling at both cysteine positions was confirmed by mass spectrometry. Astonishingly, the fluorescence intensity was reduced almost threefold

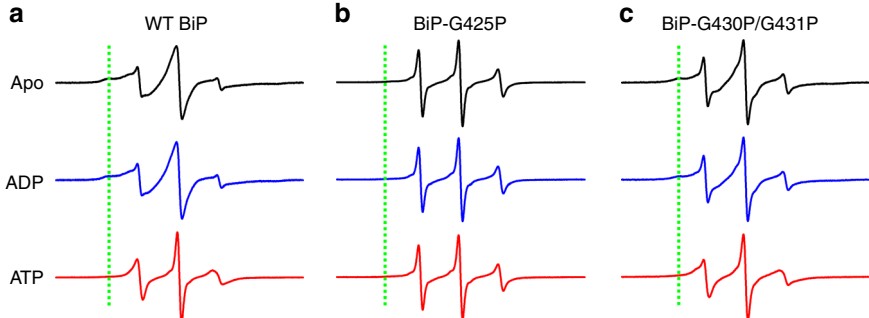

**Fig. 4** ATP binding specifically increases the dynamics of BiP's $L_{1,2}$. The purified BiP EPR construct (**a**), BiP EPR construct carrying the G425P mutation (**b**), and BiP EPR construct carrying the G430P/G431P mutation (**c**) were labeled with MTSL at V429C, and CW-EPR spectra were collected at X-band in the absence of nucleotide (Apo), or in the presence of ADP (ADP) or ATP (ATP). The green dotted lines indicate the position of the stable component if present

| | BiP-ATP2 | | BiP-ATP | | BiP-SBD | |
|---|---|---|---|---|---|---|
| | $\varphi(°)$ | $\psi(°)$ | $\varphi(°)$ | $\psi(°)$ | $\varphi(°)$ | $\psi(°)$ |
| G425 (G400) | −136.4 | 164.4 | −154.2 | −175.9 | −170.3 | −166.1 |
| G431 (G406) | −133.8 | −157.5 | 74.9 | 19.8 | 63.9 | 42.7 |

**Fig. 5** The essential role of three highly conserved glycine residues in β1-$L_{1,2}$. **a** Comparison of phi ($\varphi$) and psi ($\psi$) backbone angles for Gly425 (Gly400 in DnaK) and Gly431 (Gly406 in DnaK) in BiP-ATP2, BiP-ATP, and the isolated BiP SBD structure. **b** Fluorescence polarization assay of peptide substrate binding affinity. The model peptide substrate NR was labeled with fluorescein, and incubated with serial dilutions of BiP proteins. After binding reached equilibrium, fluorescence polarization measurements were collected. **c** The chaperone activity of DnaK in the refolding of heat denatured luciferase is compromised by G400P (G425P in BiP) and G405P/G406P (G430P/G431P in BiP) mutations. The luciferase activity before denaturation was set as 100%

after adding ATP, whereas addition of ADP has little influence (Fig. 3b, c). This strong quenching induced by ATP not only provides support for the presence of the fully closed conformation of $L_{1,2}$ in solution, but also suggests that this conformation is the major conformation for the ATP-bound state.

**ATP binding specifically increases the dynamics of $L_{1,2}$.** All available Hsp70 structures in the ADP-bound and apo states, including the isolated domain structures of SBD, have showed an almost identical conformation of the polypeptide-binding pocket: both the peptide-binding loops ($L_{1,2}$ and $L_{3,4}$) and the buttressing loops ($L_{4,5}$ for $L_{1,2}$ and $L_{5,6}$ for $L_{3,4}$) have virtually the same conformations[14–17, 20, 28, 45, 46]. This observation suggests that the conformation of the polypeptide-binding pocket is quite stable in the ADP-bound and apo states. In contrast, for all available Hsp70 structures in the ATP-bound state including BiP-ATP2, the conformations of the polypeptide-binding pocket are quite diverse. For the two DnaK-ATP structures[29, 31], they differ by the conformations of $L_{3,4}$ and $L_{5,6}$ while the flipping of β3 and β4 is

the same in both, indicating increased conformational dynamics of $L_{3,4}$ and $L_{5,6}$. This is consistent with the high B factors in this region in both structures. Previous NMR work supported the flexibility of the $L_{3,4}$ side regardless of ATP binding[45]. BiP-ATP is almost identical to one of the DnaK-ATP structures[28]. Strikingly, the BiP-ATP2 structure reported here showed an unprecedented conformation of $L_{1,2}$, and our FRET analysis supports it as the major conformation for the ATP-bound state. Together with the DnaK-ATP and BiP-ATP structures[28, 29, 31], the BiP-ATP2 structure suggests an increased flexibility of $L_{1,2}$ in the ATP-bound state. These increased dynamics are consistent with the overall increased conformational dynamics of SBDβ suggested by previous biochemical and NMR studies[18, 19, 25, 30, 35, 36].

To test whether the dynamics of BiP's $L_{1,2}$ is specifically increased in the ATP-bound state in solution, we carried out electron paramagnetic resonance (EPR) analysis. We changed Val429 on $L_{1,2}$ in BiP to cysteine, so that we could label $L_{1,2}$ with the spin label MTSSL for EPR analysis (Figs. 2a, b, 3a). The C41A and C420A modifications were also included to avoid background labeling. As shown in Supplementary Fig. 7, all the biochemical

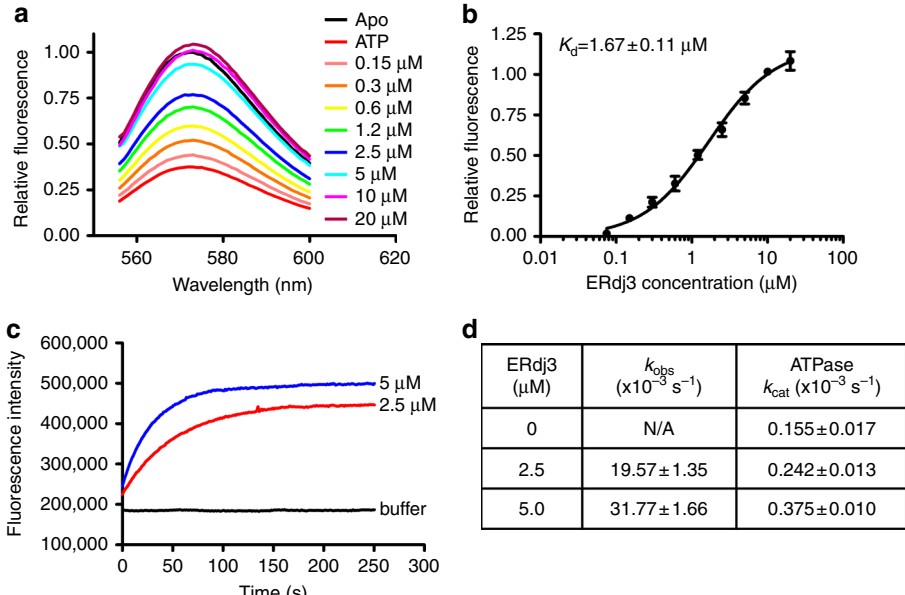

**Fig. 6** Hsp40 ERdj3 opens BiP's $L_{1,2}$ directly through binding. **a** ERdj3 significantly increased the fluorescence intensity of the TMR-labeled BiP in the presence of ATP. Serial dilutions of ERdj3 were added to the TMR-labeled BiP protein in the presence of ATP, and fluorescence emission spectra were collected with excitation at 547 nm. The concentrations of ERdj3 are labeled on the right. The spectra of Apo BiP and BiP in the presence of only ATP were used as controls. Relative fluorescence was calculated by setting the fluorescence intensity of the Apo BiP at 573 nm (the peak for TMR emission) as 1. **b** The affinity of ERdj3 for BiP. Relative fluorescence intensity at 573 nm (the peak for TMR emission) was used by setting the values of Apo and ATP alone as 1 and 0, respectively. **c** The kinetics of ERdj3-induced fluorescence increases in the TMR-labeled BiP (representing the opening of $L_{1,2}$). The fluorescence of the TMR-labeled BiP was first quenched by addition of ATP. A stopped-flow device was used to track the increase in fluorescence intensity upon mixing with indicated concentrations of ERdj3 (labeled next to the graphs). Mixing with the buffer only was used as a control (buffer). **d** The ERdj3-induced opening of $L_{1,2}$ occurs much faster than ATP hydrolysis. The rates of ERdj3-induced opening of $L_{1,2}$ ($k_{obs}$) were determined from assays done as in **c**. The ATPase rates ($k_{cat}$) were determined from single-turnover ATPase assays under the same conditions. Every rate was an average of three independent measurements

activities tested were largely intact for the BiP EPR construct, BiP-C41A/C420A/V429C, suggesting that none of the modifications has considerable effect on the biochemical activities of BiP.

Consistent with our hypothesis, in the presence of ATP, the MTSSL-labeled BiP protein has sharp peaks (Fig. 4a), supporting high conformational dynamics. By contrast, in the ADP-bound and apo states, the lines are much broader, consistent with low flexibility, potentially due to the interactions with SBDα or peptide substrates in these states as shown by the available structures[14, 15, 28].

**Conformational impact of three conserved glycine residues.** $L_{1,2}$ is dynamic and the fully closed conformation observed in BiP-ATP2 is the major conformation for the ATP-bound state. Fully closing the polypeptide-binding pocket could completely exclude substrate binding and be a crucial step in the chaperone cycle for the active release of the bound polypeptide substrates. Consistent with this hypothesis, Hsp110s such as Sse1 and human Hsp110 do not show significant binding to classic peptides for Hsp70s and exhibit little folding activity[47, 48]. To test this hypothesis, we examined the backbone conformations of β1 to β2, including $L_{1,2}$. We found that two glycine residues changed their ψ and φ peptide bond angles drastically, Gly425 in β1 and Gly431 in $L_{1,2}$, while the remaining residues were affected moderately (Fig. 5a, Supplementary Fig. 8). These two glycine residues are highly conserved in Hsp70s (Supplementary Fig. 1), and their backbone conformations are almost identical in all previous published Hsp70 structures regardless of the nucleotide-bound state (Supplementary Fig. 8), which is consistent with the virtually same conformation of $L_{1,2}$ in all these structures. Interestingly,

Ser403 in Sse1, corresponding to Gly425 in BiP, has highly similar ψ and φ peptide bond angles as BiP-ATP2, consistent with the similar conformation of $L_{1,2}$ in these two structures. Since $L_{1,2}$ in Sse1 is four-residue longer than classic Hsp70s, there is no equivalent residue for Gly431 in Sse1.

Mutating these glycine residues to proline should favor $L_{1,2}$ in the BiP-ATP2 conformation, as the resulting backbone angles would be less compatible with the projecting outward conformation of the isolated SBD and BiP-ATP structures. Consistent with this hypothesis, when the G425P mutation was introduced into the aforementioned TMR-labeled BiP protein, the fluorescence intensity for the apo state of this mutant protein was close to the labeled WT protein in the presence of ATP at the same protein concentration (Fig. 3b, d). Moreover, the ATP-induced quenching of TMR fluorescence was significantly compromised (Fig. 3c, d), suggesting that $L_{1,2}$ is most likely kept in a closed conformation by this G425P mutation regardless of the nucleotide status.

Restricting the conformation of $L_{1,2}$ in the BiP-ATP2 structure should result in reduced binding to peptide substrate and compromised chaperone activity. To test this hypothesis, we mutated these glycine residues to proline in BiP. We carried out the fluorescence polarization assay using the NR peptide to determine the peptide substrate binding activity. Supporting our hypothesis, BiP-G425P mutant protein showed little binding to the NR peptide (Fig. 5b), while WT BiP showed robust binding, as reported previously[28]. Since Gly430 is also highly conserved (Supplementary Fig. 1), we reasoned that it may contribute to the dynamics of $L_{1,2}$ like Gly431 although only moderate changes were observed for its backbone conformations among all the structures (Supplementary Fig. 8). Thus, we mutated both Gly430

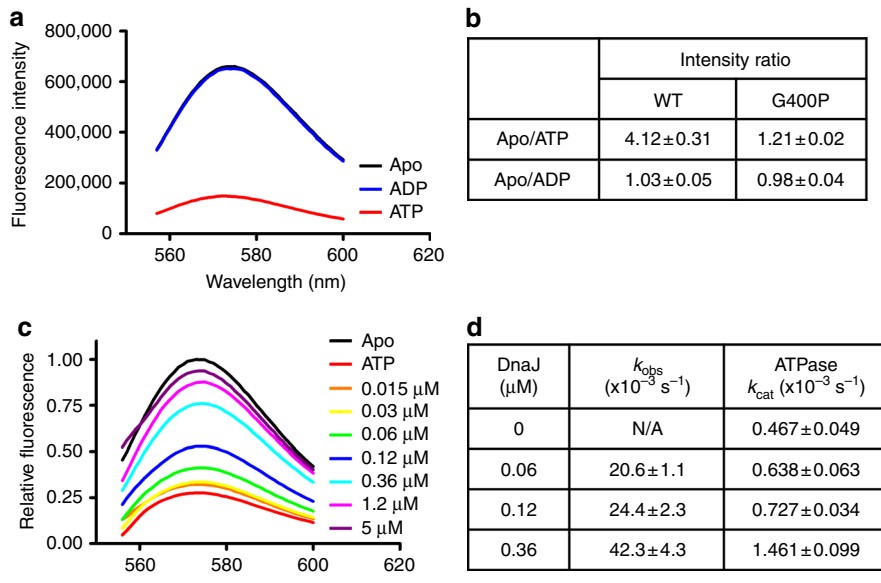

**Fig. 7** Conservation of the conformational dynamics of $L_{1,2}$ in DnaK. **a** The fully closed conformation of $L_{1,2}$ is the dominant form for DnaK in the ATP-bound state. Emission spectra of the TMR-labeled DnaK in the absence of nucleotide (Apo, black) or in the presence of ADP (blue) or ATP (red) were collected with excitation at 547 nm. **b** Fluorescence changes of the TMR-labeled DnaK in response to nucleotides. **c** Hsp40 co-chaperone DnaJ drastically increased the fluorescence intensity of the TMR-labeled DnaK in the presence of ATP. The assays were carried out as in Fig. 6a with DnaJ and the TMR-labeled DnaK. The concentrations of DnaJ were labeled on the right. The spectra of Apo and ATP alone were used as controls. **d** The DnaJ-induced opening of DnaK's $L_{1,2}$ is faster than ATP hydrolysis. Stopped-flow experiments were used to determine the rates of DnaJ-induced opening of $L_{1,2}$ ($k_{obs}$), and single-turnover ATPase assays were carried out under the same condition to determine the ATPase rate ($k_{cat}$)

and Gly431 in BiP to proline, and the binding affinity for the NR peptide was drastically reduced (Fig. 5b), consistent with our hypothesis. Since it is difficult to assay BiP's chaperone activity, we analyzed the functional impact of these proline mutations in DnaK. First, we introduced analogous proline mutations in DnaK, G400P (G425P in BiP) and G405P/G406P (G430P/G431P in BiP). Like BiP, strong defects in binding the NR peptide was observed for both DnaK mutant proteins (Supplementary Fig. 9), supporting the high conservation of these glycine residues. To assay the chaperone activity of DnaK, we carried out a refolding assay with heat denatured luciferase as a substrate. Supporting our hypothesis, severe defects were observed for both DnaK-G400P and DnaK-G405P/G406P mutant proteins (Fig. 5c). Taken together, these three highly conserved glycine residues play a pivotal role in closing $L_{1,2}$ to exclude peptide substrate binding.

Furthermore, both G425P and G430P/G431P mutations significantly compromised the ATP-induced change in the dynamics of BiP's $L_{1,2}$ (Fig. 4b, c), supporting the essential role of these glycine residues in the conformational flexibility of $L_{1,2}$. These proline mutations are supposed to restrict $L_{1,2}$ in the conformation observed in BiP-ATP2, regardless of the nucleotide-binding status. In this conformation, $L_{1,2}$ is too far away from SBDα to form contact. At the same time, it excludes peptide substrates from binding, as shown above by the drastically reduced peptide binding affinity. Thus, the increased dynamics in ADP-bound and apo states is consistent with the predicted conformation.

**Hsp40 binding opens the fully closed conformation of Hsp70.** The fully closed conformation seems to exclude binding of polypeptide substrates, providing support for an active release of bound substrates upon ATP binding. However, this conformation does not favor the binding of polypeptide substrates. Thus, these

observations raised a key question: how can Hsp70s bind polypeptide substrates efficiently in the ATP-bound state to initiate a productive chaperone cycle? To address this question, we tested the effect of the Hsp40 co-chaperone on the conformation of the polypeptide-binding pocket using the aforementioned FRET assay. As a key co-chaperone for Hsp70s, Hsp40s (also called J-proteins) have been shown to specifically interact with the ATP-bound state of Hsp70s and bring polypeptide substrates to Hsp70s to facilitate the chaperone activity of Hsp70s[1, 2, 7]. Hsp40 may be a key player in regulating the conformational landscape of Hsp70s to allow efficient substrate binding. So far, seven Hsp40 co-chaperones have been discovered for BiP[40]. Among these Hsp40s, ERdj3 is a major Hsp40 involved in protein folding, degradation, and quality control[40, 51, 52]. Thus, we added purified ERdj3 to the TMR-labeled BiP protein in the presence of ATP in order to see how the conformation of the polypeptide-binding pocket changed. As shown above, ATP drastically reduced the fluorescence intensity. Excitingly, ERdj3 significantly increased the fluorescence intensity of the TMR-labeled BiP in the presence of ATP to the level of the apo state (Fig. 6a). This observation suggested that ERdj3 shifts the fully closed conformation of $L_{1,2}$ as seen in the BiP-ATP2 structure to the open conformation of $L_{1,2}$ as observed in either the BiP-ATP structure or the isolated SBD structure (Fig. 3a). Consistent with the absence of interaction with Hsp40 for the ADP-bound and apo states of Hsp70s, ERdj3 has no appreciable effect on the fluorescence intensity in the presence of ADP (Supplementary Fig. 10). The effect of ERdj3 in the presence of ATP is concentration dependent, and fitted well with one-site binding equation with a dissociation constant $K_d$ at $1.67 \pm 0.11 \, \mu M$ (Fig. 6b). Moreover, it suggests that the fully closed conformation of the ATP-bound state is capable of interacting with Hsp40s, unlike the open conformation seen in previously published BiP-ATP and DnaK-ATP structures, which depend on the T229A mutation (T199A in DnaK) and have no detectable interaction with Hsp40s.

Our FRET assay with TMR was designed to measure the conformation of $L_{1,2}$ relative to β4, the bottom of the polypeptide-binding pocket. It does not distinguish the open conformation of the ATP-bound state from the closed conformation of the ADP-bound and apo states, since both conformations have an open $L_{1,2}$ (Fig. 3a). Thus, there are two possibilities for the ERdj3-induced opening of $L_{1,2}$. The first is that ERdj3 binding directly shifts the conformation of the polypeptide-binding pocket from the fully closed form to the open conformation in the BiP-ATP structure while BiP remains in the ATP-bound state (Fig. 3a), i.e., only $L_{1,2}$ is flipped open. The other possibility is that ERdj3 stimulates ATP hydrolysis by BiP, putting BiP in the ADP-bound state, which results in the opening of $L_{1,2}$. For the first possibility, the opening of $L_{1,2}$ should be faster than ATP hydrolysis, whereas ATP hydrolysis should occur before the opening of $L_{1,2}$ for the second possibility. To distinguish between these two possibilities, we analyzed both the speed of ERdj3-induced opening of $L_{1,2}$ and the rate of ATP hydrolysis under the same conditions. To determine the rate of ERdj3-induced opening of $L_{1,2}$, we carried out stopped-flow experiments upon addition of ERdj3. As shown in Fig. 6c, the reactions follow single-exponential kinetics. The resulting rate constants, $k_{obs}$, are listed in Fig. 6d. At the same time, we used the single-turnover ATPase assay to determine the ATP hydrolysis rates. Interestingly, the rates of ATP hydrolysis are about 80 times slower than the rates of the ERdj3-induced opening of $L_{1,2}$ for the two concentrations of ERdj3 tested (Fig. 6d). Thus, the ERdj3-induced opening of $L_{1,2}$ is most likely due to the direct effect of ERdj3 binding, way before ATP hydrolysis. Taken together, Hsp40 ERdj3 binding directly shifts BiP-ATP from the mainly fully closed conformation to the open conformation. Opening the polypeptide-binding pocket should promote Hsp70s to efficiently bind substrates in the ATP-bound state. This observation provides an explanation for the Hsp40-activated substrate binding to Hsp70s in the ATP-bound state[53, 54].

**The conformational landscape of $L_{1,2}$ is largely conserved**. Although Hsp70s are highly conserved in general, the sequences, biochemical properties, and cellular functions of eukaryotic Hsp70s such as BiP are significantly different from those of prokaryotic Hsp70s[28, 55–60]. Moreover, our previous studies have provided support for an overall conserved but differently regulated molecular mechanism of allostery among Hsp70s[28]. To understand the functional differences and conservation among Hsp70s, we analyzed the conformational dynamics of $L_{1,2}$ in DnaK, a well-studied model Hsp70 from *E. coli*. For the FRET analysis, we made analogous cysteine modifications in DnaK: M404C and T437C. Both modifications showed little impact on either DnaK's in vivo chaperone activity or its two intrinsic biochemical activities (Supplementary Fig. 11a, b). In addition, DnaK has a single endogenous cysteine, C15. To prevent background labeling, we also included a C15A modification in the DnaK FRET construct. Previous studies have shown that the C15A mutation has no obvious influence on DnaK's biochemical activities[24, 61]. Interestingly, upon addition of ATP, the fluorescence of the TMR-labeled DnaK was reduced more than fourfold (Fig. 7a, b). Thus, like BiP, the fully closed conformation of $L_{1,2}$ is not only present in solution for DnaK, but also exists as the major form for the ATP-bound state. The even stronger ATP-induced quenching in DnaK suggests that the proportion of the fully closed conformation of $L_{1,2}$ may be higher for DnaK than for BiP. Furthermore, the G400P mutation almost completely abolished this ATP-induced quenching in DnaK (Fig. 7b), and regardless of the nucleotide status, the fluorescence intensity was close to that of the WT DnaK in the presence of ATP at the same protein

concentration (Supplementary Fig. 11c), suggesting that G400P most likely locked $L_{1,2}$ in the fully closed conformation. These results are consistent with the G425P mutation in BiP, supporting the essential role of this glycine in closing $L_{1,2}$.

To test whether the intriguing regulatory effect of Hsp40 ERdj3 on the conformation of BiP's $L_{1,2}$ is conserved, we analyzed the influence of DnaJ, a well-established Hsp40 partner for DnaK[1, 62], in the FRET assay. Like ERdj3, DnaJ drastically increased the fluorescence intensity of the TMR-labeled DnaK in the presence of ATP to the level of apo state, whereas no appreciable effect was observed in the absence of ATP (Fig. 7c, Supplementary Fig. 11d). Moreover, the rates of DnaJ-induced increase in fluorescence intensity (i.e., the opening of $L_{1,2}$) is about 30 times faster than the rates of ATP hydrolysis under the same conditions (Fig. 7d, Supplementary Fig. 11f). Thus, consistent with the regulatory role of ERdj3 on the conformational landscape of BiP, DnaJ binding directly opens $L_{1,2}$ of DnaK's polypeptide-binding pocket. Remarkably, the affinity of DnaJ for DnaK deduced from the FRET analysis is within the range of previously published affinities using other approaches (Supplementary Fig. 11e)[38, 63]. Moreover, the dissociation constant of DnaJ-DnaK is about 10 times lower than that of ERdj3-BiP. This is consistent with the observation that higher concentrations of ERdj3 were required to reach similar stimulation on the ATPase activity of BiP than those of DnaJ[28].

## Discussion

The current paradigm of the Hsp70 chaperone cycle is mostly based on the differential affinities and kinetic properties of Hsp70s for polypeptide substrates in the ATP and ADP-bound states[1, 2, 15, 22, 23]. Due to the fast kinetics of the ATP-bound state, polypeptide substrate was proposed to bind to the ATP-bound state to start the chaperone cycle. Upon substrate binding, ATP hydrolysis is stimulated. Then, Hsp70 is in the ADP-bound state. With high affinity and slow kinetic for polypeptides, Hsp70-ADP holds the bound polypeptide for a period of time to prevent misfolding. Upon nucleotide-exchange, Hsp70 returns to the ATP-bound state. Due to the fast release rate, the bound polypeptide substrate is released, and Hsp70 is available to bind other substrates and restart the chaperone cycle. However, this well-accepted model did not provide an explanation for a key issue: the released polypeptide substrate is in close proximity, and may bind to Hsp70 over and over again before Hsp70s can get an opportunity to bind other substrates, which would prevent productive folding. Thus, it had been hard to explain polypeptide substrate release without rebinding. A mechanism to prevent the rebinding of the same polypeptide substrate seems important for productive folding. The fully closed conformation of the polypeptide-binding pocket reported here may provide such a mechanism by actively releasing the bound polypeptide substrates. Moreover, our FRET analysis suggests that this fully closed conformation is the major conformation of the ATP-bound state. Thus, after ATP rebinding, Hsp70 is mainly in this fully closed conformation. This conformation ensures that the bound polypeptides leave Hsp70s and will not rebind, preventing Hsp70s from repeatedly binding to the same substrate. Once the polypeptide-binding pocket opens again, other substrates will get a chance to bind. Taken together, we thus provide evidence that Hsp70s use the energy from ATP binding to actively release substrates during the chaperone cycles.

This BiP-ATP2 structure in combination with the three previously reported Hsp70-ATP structures demonstrates conformational diversity and flexibility of the polypeptide-binding pocket in the ATP-bound state. So far, all available isolated SBD structures display a nearly identical conformation for the polypeptide-binding pocket[15], suggesting the polypeptide-binding pocket in

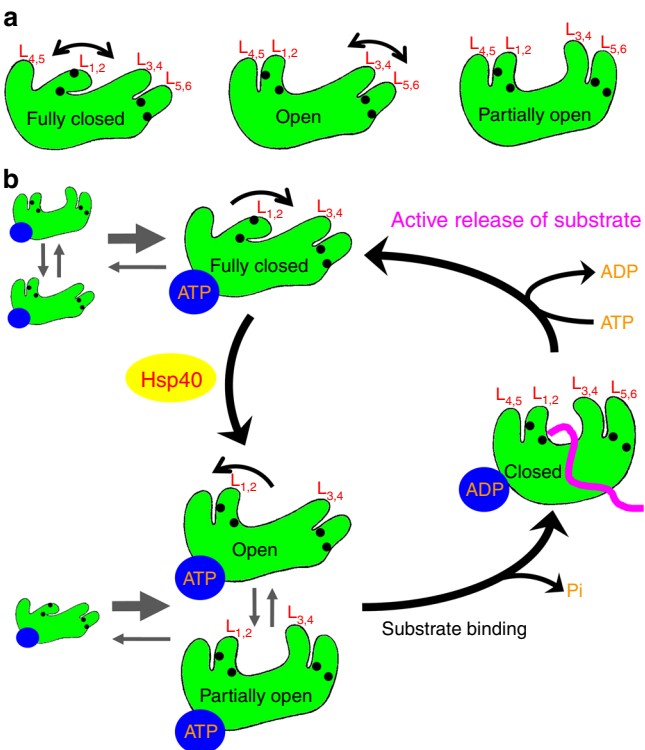

**Fig. 8** Model for the conformational diversity of the polypeptide-binding pocket and the active release of bound polypeptide substrate when Hsp70s bind ATP. **a** The conformation of the polypeptide-binding pocket is diverse and flexible in the ATP-bound state. Three conformations are shown, with only the SBDβs illustrated: fully closed (left), open (middle), and partially open (right). The two peptide-binding loops ($L_{1,2}$ and $L_{3,4}$) and their supporting loops ($L_{4,5}$ and $L_{5,6}$) are labeled. The black circles represent the highly conserved glycine residues in $L_{1,2}$ and $L_{5,6}$ that function like joints. **b** The Hsp70 chaperone cycle with an active release of bound substrate and Hsp40 as the key for regulating substrate binding. As in **a** only the SBDβs are shown. The bound-nucleotides are shown as blue ovals, and the bound polypeptide substrate in the ADP-bound state is highlighted in purple

the ADP-bound and apo states is quite rigid. The conformation of the polypeptide-binding pocket in the ATP-bound state is thus more dynamic than in the ADP-bound and apo states, which is supported by our EPR analysis and a number of previous studies[18, 19, 25, 30, 35, 36]. However, crystal structures may have captured only the extreme cases, and it is possible that additional conformations of the polypeptide-binding pocket are present. The closed conformation in isolated SBD may also exist in the ATP-bound state, but the population may be low.

Our BiP-ATP2 structure and biochemical analysis suggests that the two peptide-binding loops are quite flexible and dynamic in the ATP-bound state, consistent with previous studies[25, 35]. The three highly conserved glycine residues in the $\beta1$-$L_{1,2}$-$\beta2$ region and the two highly conserved glycine residues on $L_{5,6}$ reported previously[28, 29] may serve as joints to allow the flexibility to these loops (Fig. 8a). The polypeptide-binding pocket is like a hand, and the two peptide-binding loops are analogous to our fingers. This flexibility of the peptide-binding loops allows Hsp70s to bind and release peptide substrate efficiently. In the ATP-bound state, this "hand" can assume a number of different conformations (Fig. 8a): completely open as in the DnaK-ATP structure reported previously by our group (open), partially open as in the DnaK-ATP structure reported by the Mayer group (partially open), and completely closed as in the BiP-ATP2 structure reported here (fully closed). The closed conformation with a

bound substrate as in the ADP-bound state (closed, Fig. 8b) may also exist at very low level. Thus, in the ATP-bound state, these different conformations are in constant equilibrium with the fully closed conformation as the major form. The fast kinetics and low affinity for peptide substrates observed in biochemical assays are a result of the combination of all these conformations.

How can the ATP-bound state bind polypeptide substrate efficiently to initiate a productive chaperone cycle? Our FRET studies show that Hsp40 binding directly shifts the fully closed conformation to the open conformation for Hsp70s in the ATP-bound state. This apparent regulatory role of Hsp40 is conserved between ERdj3 and DnaJ although ERdj3 was shown to function as a tetramer[64] while DnaJ functions as a dimer[1, 7]. This Hsp40 effect thus promotes Hsp70s to efficiently bind polypeptide substrates to initiate productive chaperone cycles. This is consistent with the essential role of Hsp40s in assisting the chaperone activity of Hsp70s in protein folding[1, 2, 7]. This role of Hsp40s provides additional evidence for the proposed function of Hsp40s in bringing polypeptide substrates to Hsp70s.

Taken together, we propose a refined model of the Hsp70 chaperone cycle (Fig. 8b). In the ATP-bound state, the polypeptide-binding pocket of Hsp70s can adopt a number of conformations as described above with the fully closed conformation as the major form (Fig. 8b, top left). Thus, Hsp70-ATP by itself has very low ability in binding polypeptide substrates. Hsp40 co-chaperone is the key to initiating a productive chaperone cycle, by shifting the equilibrium toward open conformations (Fig. 8b, bottom left). Now, Hsp70s can efficiently bind polypeptide substrates to initiate a new, productive chaperone cycle. With both Hsp40 and substrate bound, ATP hydrolysis occurs and Hsp70s are converted to the ADP-bound state (Fig. 8b, right). The polypeptide-binding pocket is closed on to the substrate, and Hsp40s are released from Hsp70s. Upon nucleotide exchange to ATP, Hsp70s are back to the ATP-bound state. The fully closed conformation becomes the dominant conformation again, and it propels the release of the bound substrate. This active release of bound substrate provides an opportunity for the released substrate to refolding, and prevents Hsp70s from binding to the same substrate over and over again, which could result in non-productive chaperone cycles. Once Hsp40 binds, a new productive chaperone cycle is re-initiated.

Questions remain regarding what controls the conformation of the polypeptide-binding pocket. Does the polypeptide-binding pocket simply oscillate among the different conformations or is there a signal for it to assume a specific conformation? Interestingly, the β8 of SBDβ also has a diverse conformation (Fig. 2c and Supplementary Figs. 4b, 6) and has been shown to be essential for allosteric coupling. Mild mutations on β8 have caused chemical shifts at Gly400 in DnaK, one of the highly conserved glycine residues in $\beta1$-$L_{1,2}$ region[35]. It is possible that β8 may have a role in influencing the conformation of the polypeptide-binding pocket. In all three previously reported Hsp70-ATP structures, the polypeptide-binding pocket assumes open conformations, either fully open or partially open. All of these structures depend on the ATPase deficient T199A mutation in DnaK, which completely abolishes Hsp40 interaction and chaperone activity. In contrast, our FRET analysis with DnaJ suggested that the fully closed conformation of the polypeptide-binding pocket is capable of interacting with Hsp40. Thus, it is possible that the T199A mutation favors the open conformation for the polypeptide-binding pocket. Although the overall conformation of NBD is highly similar between the BiP-ATP2 structure, which has a fully WT NBD, and other Hsp70-ATP structures, subtle changes due to the lack of the side-chain of threonine at 199 could shift the conformational equilibrium for the polypeptide-binding pocket. Another key question is: how does Hsp40 change the

conformational equilibrium for the polypeptide-binding pocket of Hsp70s? An Hsp70–Hsp40 complex structure will be helpful to elucidate a molecular mechanism.

## Methods

**Protein expression and purification**. The crystallization construct BiP-L$_{3,4}'$ (residues 25–633) was cloned into a pSMT3 vector (a generous gift from Dr. Chris Lima), and expressed as a Smt3 fusion protein with a His$_6$ tag at the N-terminus in BL21(DE3) Gold. This construct carries a previously characterized L$_{3,4}'$ modification: the L$_{3,4}$ is replaced with a shortened sequence (TASDNQP → VGG)[28]. The induction was done at 30 °C for 6 h with 1 mM IPTG. After breaking open the cells with sonication, the fusion protein was first purified on a HisTrap column using 2×PBS buffer (20 mM Na$_2$HPO$_4$, 1.76 mM KH$_2$PO$_4$, 274 mM NaCl, and 5.4 mM KCl). The Smt3 tag together with the His$_6$ tag was removed by incubating with Ulp1 protease during an overnight dialysis in 2×PBS. The resulting BiP-L$_{3,4}'$ protein was separated from Smt3 tag on a second HisTrap column equilibrated with 2×PBS, and further purified using HiTrap Q with 50–600 mM NaCl gradient in buffers containing 25 mM Hepes-KOH, pH 7.5, and 1 mM DTT. The final step of the purification is Superdex 200 16/60 columns using buffer containing 25 mM Hepes-KOH, pH 7.5, 150 mM NaCl, and 1 mM DTT. All the columns are from GE Healthcare Life Sciences. The purified BiP-L$_{3,4}'$ protein was concentrated to ~30 mg ml$^{-1}$ in a buffer containing 5 mM Hepes-KOH (pH 7.5) and 10 mM KCl, and flash frozen in liquid nitrogen.

All the BiP proteins used for biochemical, FRET, and EPR assays were cloned, expressed, and purified essentially the same way as the BiP-L$_{3,4}'$ protein. The plasmid for expressing ERdj3 protein was a generous gift from Dr. Linda Hendershot. The expression and purification of ERdj3 was the essential the same as described before[28, 65]. Briefly, His-tagged ERdj3 was expressed in M15 E. coli with 0.1 M IPTG, and purified on a HisTrap column with 2 M urea in the lysis buffer. After the HisTrap column, the purified ERdj3 protein was dialyzed against a buffer containing 25 mM Hepes-KOH, pH 7.5, 200 mM NaCl, 20% glycerol, and 0.02% Triton X-100, and flash frozen in liquid nitrogen before storing in −80 °C freezer.

All the DnaK proteins were expressed and purified as described previously[29, 66]. Briefly, all the DnaK proteins were cloned into the *dnak* expression plasmid pBB46 with a His$_6$ tag at the C-terminus, and expressed in the *dnak* deletion strain BB205[67]. After a HisTrap column with 2×PBS buffer, all the DnaK proteins were further purified on a HiTrap Q column using buffers containing 25 mM Hepes-KOH, pH 7.5, and 1 mM DTT. The DnaJ protein was purified as described before[28, 66]. Briefly, DnaJ was cloned into pSMT3 vector, and expressed as Smt3-DnaJ fusion protein at 30 °C using BL21(DE3) Gold. The Smt3-DnaJ fusion protein was first purified on a HisTrap column using buffers containing 25 mM Hepes-KOH, pH 7.5, and 300 mM KCl. The Smt3 tag was cleaved by Ulp1 protease and removed by a second HisTrap column. DnaJ protein was further purified using a Superdex 200 16/60 column equilibrated with a buffer containing 25 mM Hepes-KOH, pH 7.5, 300 mM KCl, and 1 mM DTT. All the purified proteins were concentrated to >10 mg ml$^{-1}$, flash frozen in liquid nitrogen and stored in −80 °C freezer.

**Structure determination**. Before setting up crystallization trials, the purified BiP-L$_{3,4}'$ protein was diluted to 10 mg ml$^{-1}$ in a buffer containing 5 mM Hepes-KOH (pH 7.5), 10 mM KCl, 5 mM Mg(OAc)$_2$, and 2 mM ATP. Crystals of BiP- L$_{3,4}$ were grown at 20 °C using a hanging-drop vapor diffusion method with a mother liquor containing 18–22% PEG 1000, 0.1 M phosphate citrate (pH 4.2–7.0), and 0.2 M Li$_2$SO$_4$. Since initial crystals were plate-clusters, micro-seeding was used to obtain single crystals with a mother liquor containing 12–15% PEG 1000, 0.1 M phosphate citrate (pH 4.2–7.0), 0.2 M Li$_2$SO$_4$, and 2% (w/v) dioxane. Single crystals were looped out, cryo-protected with 15% MPD in mother liquor, and flash frozen in liquid nitrogen. SeMet protein of BiP-L$_{3,4}$ was prepared, and crystals were obtained the same way as the native protein.

Two diffraction data sets were collected at Beamline X4C of the Brookhaven National Laboratory: a native data set at 1.85 Å and Se-SAD data set at 2.0 Å. Data indexing, integration, and scaling were carried out in HKL2000[68]. Phases were evaluated on the Se-SAD data set with hkl2map[69], and initial model was obtained by automated model building with Arp/wArp[70]. Model building and refinement were carried out with COOT[71] and Phenix[72] using the native data set. The resulting model has excellent Ramachandran statistics: out of total 1271 residues, 1237 residues (97.3%) are in favored regions, and 1270 residues (99.9%) are in allowed regions (MolProbity Ramachandran analysis).

**Site-directed mutagenesis and growth tests**. All BiP and DnaK mutations were obtained with QuikChange Lightening site-directed Mutagenesis kit (Stratagene). Growth tests of E. coli on DnaK were performed as described previously[27, 66, 67]. Briefly, all the DnaK mutants were cloned into the aforementioned *dnak* expression plasmid pBB46, and transformed into the *dnak* deletion strain BB205. Growth tests were done with fresh transformations, and on LB plates containing 50 μg ml$^{-1}$ ampicillin, 25 μg ml$^{-1}$ kanamycin, 25 μg ml$^{-1}$ chloramphenicol, and 20 μM IPTG with one-overnight incubation at 37 °C.

**TMR labeling and FRET analysis**. BiP and DnaK proteins used for TMR labeling were purified to high purity as described above in the presence of 1 mM DTT. After fully reduction with 5 mM DTT for 2 h on ice in degassed buffer A (30 mM Tris-HCl, pH 7.5, 25 mM KCl, and 10% glycerol), DTT was removed by PD10 desalting column (GE Healthcare Life Sciences) equilibrated with buffer A. Protein concentration was adjusted to 5 mg ml$^{-1}$. Tetramethylrhodamine-5-iodoacetamide (5-IATR, AnaSpec Inc.) was dissolved in dry DMF at 6.5 mM. DnaK proteins were mixed with 5-IATR at 1:4.5 molar ratio, and incubated in the cold-room overnight with end-over-end stirring to get complete labeling. After the reaction was stopped by adding 2 mM sodium 2-mercaptoethanesulfonate, the excess 5-IATR label was removed by PD-10 column equilibrated with buffer A. The labeled protein was further purified on a 1 ml HiTrap Q column (GE Healthcare Life Sciences) with 25–1000 mM KCl gradient in buffers containing 30 mM Tris-HCl, pH 7.5, 3 mM MgCl$_2$, and 10% glycerol. The peak corresponding to monomer was used for FRET analysis. Complete labeling at both introduced cysteine positions were confirmed by mass spectrometry.

FRET analysis was carried out at room temperature on a PC1 Photon Counting Spectrofluorimeter from ISS Inc. (IL, USA). Buffer B (30 mM Tris-HCl, pH 7.0, 100 mM KCl, 3 mM MgCl$_2$, 3 mM CaCl$_2$, 10% glycerol) and buffer C (30 mM Tris-HCl, pH 7.5, 25 mM KCl, 3 mM MgCl$_2$, and 10% glycerol) were used for BiP and DnaK, respectively. For the nucleotide-effect experiments, BiP and DnaK proteins were diluted to 300 nM. Emission spectra were collected from 556 to 600 nm with excitation at 547 nm. For the samples with ATP, ATP was added to a final concentration of 2 mM and incubated for 2 min for DnaK and 10 min for BiP to allow binding right before reading. ADP was added at a final concentration of 100 μM for the samples in the presence of ADP.

For the Hsp40 effect experiment, TMR-labeled BiP and DnaK proteins were diluted to final concentrations of 300 and 30 nM, respectively, based on stability. ATP was added to a final concentration at 2 mM, and incubated for 2 min for DnaK and 10 min for BiP to fully quench the fluorescence. Then, purified ERdj3 or DnaJ protein was added at the indicated concentrations. After incubating for 2–5 min to allow binding to reach equilibrium, emission spectrum was collected with excitation at 547 nm. For the Apo control, ATP was not added.

The stopped-flow experiments were carried out using a HiTech SFA-20 Rapid Kinetics accessory instrument. TMR-labeled BiP and DnaK (60 nM) were first incubated with 2 mM ATP to quench the fluorescence. Hsp40 ERdj3 and DnaJ proteins were injected as indicated and the changes in fluorescence intensity were tracked over time. All the binding curves fitted well with the one-phase exponential association and association constant, $k_{obs}$, were deduced using Prism (GraphPad).

**Peptide substrate binding assay**. The NR peptide was labeled with fluorescein at the N-terminus and ordered from NEOBioscience (at >95% purity). Binding affinity assays were carried out as described previously with some modifications[28, 29, 48, 66]. Briefly, serial dilutions of Hsp70 proteins were prepared in buffer D (25 mM Hepes-KOH, pH 7.5, 100 mM KCl, and 10 mM Mg(OAc)$_2$, and 1 mM DTT) containing 10% glycerol, and incubated with NR peptide at a final concentration of 10 nm. After the binding reaches equilibrium, fluorescence polarization measurements were carried out on Beacon 2000 (Invitrogen) and dissociation constants ($K_d$) were calculated using Prism (GraphPad).

**Single-turnover ATPase assay**. The assay was carried out as described previously with some modifications[28, 66]. Briefly, 2 μg of purified BiP and DnaK proteins was diluted with buffers B and C, respectively, and then incubated with 2 μl of [α-32P] ATP (NEG503H250UC, 3000 Ci/mmol; Perkin Elmer) in a final volume of 20 μl on ice for 2 min to allow ATP binding to these Hsp70 proteins. The Hsp70–ATP complex was quickly isolated from free ATP on a spin column pre-equilibrated with buffer B for BiP and Buffer C for DnaK, aliquoted and frozen in liquid nitrogen. Then, the complex was diluted to about 60 nm, and each reaction was started by mixing equal volumes of the Hsp70–ATP complex with either ERdj3 or DnaJ at the indicated concentrations. After incubating at 22.5 °C, the reactions were stopped at indicated time points with stop buffer (1 M formic acid, 0.5 M LiCl, and 0.25 mM ATP). An aliquot of 1.5 μl for each reaction was spotted on PEI-cellulose thin-layer chromatography plates (Sigma-Aldrich) to separate ATP from ADP. After the amount of radioactive ATP and ADP were visualized and quantified with a Typhoon phosphorimaging system (GE Healthcare), the rate of ATP hydrolysis ($k_{cat}$) was calculated using a first-order rate equation by nonlinear regression with Prism (GraphPad).

**EPR analysis with site-directed spin labeling**. All the purified BiP EPR proteins were stored in a buffer containing DTT to prevent cysteine residues from oxidation. Right before MTSL (R1: 1-oxyl-2,2,5,5-tetramethyl-D3-pyrr-oline-3-methyl-methanethiosulfonate, Toronto Research Chemicals) labeling, DTT was removed using a PD10 desalting column (GE Healthcare) equilibrated with buffer E (25 mM Hepes-KOH, pH 7.0, 100 mM KCl, and 10 mM Mg(OAc)$_2$). BiP proteins were mixed with MTSL (dissolved in acetonitrile) at 1:5 molar ratio, and incubated at 4 °C overnight in dark. Free MTSL was removed from the labeled proteins on a second PD-10 column. The labeling efficiency was over 90%. The labeled proteins were concentrated to over 10 mg ml$^{-1}$. Each MTSL-labeled BiP protein was diluted to 100 μM in buffer E (pH 7.5). For samples with ADP and ATP, ADP and ATP

were added to final concentrations of 200 μM and 2 mM, respectively, and incubated for 1–2 min. A 10 μl sample was loaded into a glass capillary. Continuous-wave EPR (CW-EPR) spectra at X-band (9.849 GHz) were collected on a Bruker Spectrometer (Bruker Biospin GmbH, Rheinstetten, Germany) at 2 mW over a sweep width of 100 Gauss with amplitude of 1 Gauss at a modulation frequency of 100 kHz. To obtain reasonable signal/noise ratio, 20–50 scans were collected for each sample.

**Luciferase refolding assay**. The assay was carried out as described previously with some modifications[29]. Purified firefly luciferase (purchased from Promega) was diluted in buffer D in the presence of 3 mM ATP, and heat denatured at 42 °C. A reaction mixture containing 3 μM DnaK, 0.67 μM DnaJ, and 0.33 μM GrpE in buffer A was added to the denatured luciferase to start refolding. Luciferase activity was read in a Berthold LB9507 luminometer after incubating for indicated periods of time at room temperature.

**Data availability**. Atomic coordinates and structure factors have been deposited in the RSCB Protein Data Bank under the accession number 6ASY. The data that support the findings of this study are available from the corresponding author upon request.

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

## Acknowledgements

We thank Drs. Elizabeth Craig, Lois Greene, Clive Baumgarten, Margaret Biber, Diomedes Logothetis, and Wei Yang for critically reading the manuscript and providing insightful suggestions. We are grateful to staff at Brookhaven National Laboratory Beamline X4A and X4C, Qun Liu, John Schwanof, Randy Abramowitz, and Xiaochun Yang, for their assistance in collecting diffraction data. We thank Drs. Carlos Escalante and Vishaka Santosh for help with the stopped-flow work, and Drs. Martin Webb and Taekjip Ha for suggestions on FRET analysis. We thank Drs. Xinping Xu and Melesse Nune for technical support and discussion, Pranav Bommineni for critically reading our manuscript, and Marissa Kieber and Tsega Solomon for help with EPR analysis. This work was supported by NIH (R01GM098592 to Q.L.), Blick Scholar Award from VCU (to Q.L.), and American Heart Association (17GRNT33660506 to Q.L.). L.Z. is partially supported by 1R01GM109193 from NIH.

## Author contributions

Q.L. designed the study and wrote the manuscript. J.Y. performed most of the experiments. Q.L., J.Y. and L.Z. analyzed the structures, and carried out data analysis. All authors read the manuscript and made revisions.
