## [Peer Review File · Nature Communications]

Reviewers' comments:

Reviewer #1 (Remarks to the Author):

Yang et al. report a new x-ray crystal structure of human BiP, the endoplasmic reticulum Hsp70 chaperone, bound to ATP. The structure is of high quality (resolution 1.85 Å) and presents a novel architecture for the substrate-binding domain, in that a loop that normally extends into solvent is folded back and covers the substrate-binding cleft. The authors posit that this novel structure is relevant to the mechanism of substrate release upon ATP binding to BiP. They present complementary data from mutagenesis of the E. coli homologue to BiP, DnaK, that supports the importance of two glycine residues in conferring mobility to the loop that covers the binding site, and also use spin labeling and EPR of Cys mutants of both DnaK and BiP (endogenous Cys were mutated to Ala) to show that the loop becomes more mobile in the presence of ATP.

The writing, presentation of data, and description of methods in this paper are good and complete. However, there are fundamental logic flaws that detract from the overall thesis:

1. The BiP version used is a mutant with a shortened loop 3,4. The authors state that this protein was characterized previously, but at least here they need to state whether it is active and how its activity differs from wild type. Drawing general conclusions from this mutant seems questionable.
2. The idea that one can see different x-ray crystal forms and use them to deduce dynamics of conformational change that underlie mechanism is hard to swallow. These structures are static and populated in crystals with packing interactions. For sure the packing interactions must be shown and described fully, but even then, calling the different arrangements of the substrate-binding domain in different crystals snapshots of conformers populated during the mechanism, without validation from other data, is a stretch.
3. The switch to DnaK for some of the experiments is poorly justified. Was this done because BiP is not amenable to biochemical characterization?
4. The EPR experiment required two mutations: changing the native Cys to Ala, and introducing the new Cys. I didn't see any characterization of the function of these proteins.

Thus, this paper is not acceptable as is, and could be improved. But if the data are not used to propose a mechanism of allostery, which seems a stretch, then it is not clear how broadly read it will be. It may not be appropriate for Nature Communications, even if revised.

Reviewer #2 (Remarks to the Author):

The manuscript by Yang et al. describes the crystal structure of a novel ATP-bound state of the human Hsp70 protein BiP. In contrast to previous Hsp70-ATP structures, the authors did not employ an ATPase deficient mutant of the nucleotide binding domain (NBD), but used the strategy of deleting a loop in the substrate binding domain (SBDbeta), L3,4, to obtain the ATP-bound, open conformation. Using this variant, the authors obtained a BiP structure that seems to represent a novel ATP-bound state, in which SBDbeta is closed, and probably not able to bind substrate polypeptides. Biochemical and EPR experiments are used to show the active release of substrates following ATP binding and the dynamics of the SBD.

Overall, the manuscript is of good quality and the new BiP conformation obtained by the authors seems to underline the structural flexibility of Hsp70s and adds to the understanding of Hsp70 chaperones.

However, there are a number of points that need to be addressed by the authors:

The L3,4 modification prevents dimerization of BiP and affects substrate binding of Hsp70s (see e.g. Rist et al. 2006). Therefore, this variant is a matter of controversy in the field as to what extend it can be considered as an intact Hsp70. The authors should discuss this in the manuscript. How do the authors exclude that the novel conformation is due to using this variant? Along these lines, the BiP L3,4 variant should be included in the fluorescence anisotropy assay (Fig. 4b) to provide a better correlation of the structure with the biochemical data.

A complete structure based sequence alignment should be added to the supplement.

The X-ray structure seems overall of good quality – based on the data collection and refinement statistics. However, when reading the validation report, there are a few issues that the authors have to explain:

The structure was refined using Phenix and Refmac, but the geometries have not been properly restrained as the bond length variation and the angles are by far too high. Please explain. Along this line - the authors should provide the Molprobit output.

The $I/\sigma I$ is 2.6, which indicates that the crystals diffract to higher resolution than the cutoff used by the authors. The data should be cut according to the $cc1/2$ criteria (Karplus and Diederichs,

Science 2012). The authors should provide a $cc1/2$ value to explain this discrepancy. The redundancy of the structure is low, probably due to the low symmetry (P1). Merging different crystals or using separately collected high and low resolution datasets might be an option to improve the data - if available.

The methods section is rather superficial, lacking important details. This includes buffers used in the purification, the yeast strains used in the growth assays including Ssa (or at least proper citations). Likewise, the citations for crystallographic software are completely absent (COOT, Phenix, ARP/wArp,...). This is a bad habit as the funding of the developers depends on the citations and the citations should be added.

It is not clear how the figures were generated (Pymol??).

Figure 2 d-f

Which kt was used to calculate the surfaces of the SDBbeta? Please also cite APBS if used to calculate the surfaces.

Minor points:

The crystal packing shown in suppl. Fig. 1f (mentioned in the main text line 104) should be extended to clarify whether SDBbeta is involved in crystal contacts.

In the discussion (line 293) a citation should be included as the dynamics of the SBD with respect to ATP binding has been described for DnaK in Rist et al (2006).

The figures are labeled in different fonts which are sometimes hard to read (e.g. Fig. 1a,b) and the position of the labels needs improvement (e.g. Fig 5, II, L3,4/L5,6 label in the figure).

The model provided in Fig. 5 is of poor quality and needs improvement.

Supplementary Fig. 5b:

The proper abbreviation for molecular weight should be used (kDa).

We would like to thank the reviewers for their constructive and insightful comments to improve our manuscript. We have made every effort to address the concerns thoroughly. In response to the reviewers' comments, we have made the following changes to our manuscript. We hope that we have addressed all of the issues raised by the reviewers, significantly improved our manuscript, and made our manuscript suitable for publication on *Nature Communications*.

A. Reviewer 1:

Overall, this reviewer thinks that our new X-ray crystal structure of human BiP “is of high quality (resolution 1.85 Å) and presents a novel architecture for the substrate-binding domain”, which was supported by our complementary data from mutagenesis and EPR, and “The writing, presentation of data, and description of methods in this paper are good and complete.” At the same time, the reviewer raised the following concerns (in blue fonts). Below, we address the reviewer's concerns point-by-point (in black fonts).

1. *1), The BiP version used is a mutant with a shortened loop 3,4. The authors state that this protein was characterized previously, but at least here they need to state whether it is active and how its activity differs from wild type.*

2), *Drawing general conclusions from this mutant seems questionable.*

Response and Change:

1) We thank the reviewer for pointing this out and apologize for our oversight. As the reviewer suggested, we have added the description of this mutation in the revised manuscript. Please see the second paragraph in the RESULTS and Supplementary Fig. 3. Briefly, this modification does not affect the structural integrity of BiP although it has reduced affinity for peptide substrates.

2) We agree with the reviewer that simply drawing general conclusions from this mutant is premature. This is the reason that we carried out mutagenesis and EPR analysis to test and support the novel conformation observed in this new structure in our original manuscript, and now we have provided further support from FRET studies (please see the section “**The unique conformation of L_{1,2} observed in the BiP-ATP2 structure is present as a dominant form in solution**” in the revised manuscript).

a) We agree that mutations may change the conformation landscape of proteins. However, they do represent specific conformations that are stabilized by the mutations, and the structures of mutant proteins are extremely useful in understanding structures and dynamics of proteins especially when the wild-type proteins fail to crystallize. In the case of Hsp70s, no WT protein has been able to crystallize. So far, all the published full-length Hsp70 structures in the ATP-bound state strictly depend on the presence of two mutations in the same construct: 1) the ATPase-deficient T199A mutation in NBD, and 2) the introduced disulfide bond to link NBD and SBD (based on the Sse1 structure) or the L_{3,4} mutation in SBD used in our structures. In fact, our BiP-ATP2 structure has the least mutations among all the available Hsp70-ATP structures. It only contains the L_{3,4} modification with a completely WT NBD (i.e., it does not contain the T199A mutation). The T199A mutation not only knocks out chaperone activity but also completely abolishes Hsp40 interaction. It is still unclear which step of the ATP-bound state this T199A mutation represents.

b) As stated above, to further test whether the novel conformation of the polypeptide-binding pocket observed in our new BiP-ATP2 structure exists in solution and is not due to the L_{3,4} modification, we

have carried out FRET analysis in conjunction with our earlier mutagenesis and EPR analysis. As described in our revised manuscript, we have used tetramethylrhodamine (TMR) to label both L_{1,2} and the bottom of the polypeptide-binding pocket (β 4) where L_{1,2} closes upon to. Upon ATP binding, strong quenching of TMR was observed, suggesting ATP binding causes L_{1,2} to move closer to the bottom of the polypeptide-binding pocket, consistent with the fully-closed conformation in the BiP-ATP2 structure. Moreover, the G400P mutation almost abolished this ATP-induced quenching. Taken together, these data support that the fully-closed conformation of L_{1,2} observed in our BiP-ATP2 structure exists for WT Hsp70s as a major form in solution, and G400 is critical for this conformational change. Please see detailed description in the revised manuscript, section “**The unique conformation of L_{1,2} observed in the BiP-ATP2 structure is present as a dominant form in solution**” in the RESULTS.

2. The idea that one can see different x-ray crystal forms and use them to deduce dynamics of conformational change that underlie mechanism is hard to swallow. These structures are static and populated in crystals with packing interactions. For sure the packing interactions must be shown and described fully, but even then, calling the different arrangements of the substrate-binding domain in different crystals snapshots of conformers populated during the mechanism, without validation from other data, is a stretch.

1) The idea that one can see different x-ray crystal forms and use them to deduce dynamics of conformational change that underlie mechanism is hard to swallow.

Response and Change:

a) As described in our manuscript and above, our construct is different from all the other constructs in previously published Hsp70-ATP structures. This is the basis for different crystal forms. Up to now, four structures of Hsp70s in complex with ATP have been published: two DnaK, one BiP and one Ssb. Besides either the engineered disulfide bonds between NBD and SBD or the L_{3,4} modification, all these published structures strictly depend on a mutation in NBD that significantly reduces ATP hydrolysis, T199A for DnaK or T229A for BiP. However, this T199A mutation not only knocks out chaperone activity but also completely abolishes Hsp40 interaction. It is unclear which step of the ATP-bound state it represents.

Our BiP-ATP2 structure is unique: it does not contain this ATPase-deficient T199A mutation, i.e., it has a completely wild-type NBD. Thus, it is very possible that the BiP-ATP2 structure assumes a different conformation that represents a different step in the ATP-bound state. In fact, our FRET studies suggested that the fully-closed conformation of L_{1,2} observed in our BiP-ATP2 structure is the major conformation in solution for the ATP-bound state.

Moreover, as pointed above, the T199A mutation abolishes both chaperone activity and Hsp40 interaction. In contrast, the conformation in our BiP-ATP2 structure is capable of interacting with Hsp40 as suggested by our FRET analysis (please see Fig. 6). Thus, this T199A mutation most likely locks Hsp70s in a conformation that is different from the BiP-ATP2 structure. Indeed, in all the published structures relying on this T199A mutation, the polypeptide-binding pocket essentially assumes the same conformation no matter what the rest mutations required for crystallization are and regardless of crystal contacts (please see Supplementary Fig. 5 for crystal contacts in the two DnaK-ATP structures).

b) In a broad sense, this is a question that every crystallographer and scientist who uses crystal structures in their research has to face every day.

First, it has been well-established that “one can see different x-ray crystal forms” for a single protein. This has been well-supported by the presence of multiple structures of the same protein in the PDB database including hundreds of structures of T4 lysozyme, Streptavidin, and cutinase.

Secondly, as the reviewer pointed out, it is well-established that proteins are dynamic. They constantly sample multiple conformations. An individual crystal structure captures a unique conformation stabilized under that specific crystallization condition, a static structure. Just to clarify, in general, crystallization only stabilizes or “catches” one of the pre-existing conformations in solution. This has been well supported by a number of studies.

Thirdly and most importantly, numerous studies have supported that structures obtained under different conditions are snapshots of protein dynamism and have revealed important mechanisms. Here are some references: Lange et al., *Science* (2008) 320:1471-1475; Kondrashov et al., *Proteins* (2008) 70:353-362; Burmann et al., *Cell* (2012) 150:291-303; Kossiakoff et al., *Proteins* (1992) 14:65-74; Juritz et al., *Nucleic Acid Research* (2011) 39:D475-9; and Burra et al., *PNAS* (2009) 106:10505-10510. An extreme example is time-resolved crystallography. A number of beautiful works were carried out using time-resolved X-ray crystallography on a single crystal to reveal amazing conformational changes and mechanisms (such as Schotte et al., *Science* (2003) 300:1944-1947, Nakamura et al., *Nature* (2011) 487: 196-201).

c) We fully agree that only structure is not enough especially when a structure is stabilized with a mutation. However, as pointed out above in Response for Concern #1, we have comprehensive and solid solution studies to support our structural observations. We started with a new crystal structure of BiP in the presence of ATP. This new structure generated hypothesis for us to test. Importantly, our solution studies including FRET, biochemical, and EPR data provided strong support for the hypothesis from the structural observation.

2) *These structures are static and populated in crystals with packing interactions. For sure the packing interactions must be shown and described fully, but even then, calling the different arrangements of the substrate-binding domain in different crystals snapshots of conformers populated during the mechanism, without validation from other data, is a stretch.*

Response and Change:

a) We agree with the reviewer that each crystal structure is “static and populated in crystals with crystal packing interactions”. As the reviewer suggested, we have shown crystal packing interactions in our new BiP-ATP2 structure with detailed description (please see the second to the last paragraph of the section “**A surprising new conformation of the polypeptide-binding pocket**” in the revised manuscript). In fact, there is only one hydrophobic contact between L_{1,2} and the symmetry mates. As comparison, we also have included the crystal packing interactions around the peptide-binding loops for the two previously published DnaK-ATP structures. L_{1,2} in these two structures actually participates more extensive crystal contacts than that of our BiP-ATP2 structure. Moreover, L_{1,2} assumes virtually identical conformation in these two DnaK-ATP structures in spite of completely different crystal contacts, supporting that crystal contacts have little influence on the conformation of this loop. This is consistent with the aforementioned concept that crystallization only stabilizes or “catches” one of the pre-existing conformations in solution.

b) We also agree with the reviewer: simply using “the different arrangements of the substrate-binding domain in different crystal snapshots conformers populated during the mechanism, without validation from other data, is a stretch”. However, as the reviewer pointed out in his/her summary at the beginning of his/her review, we did provide complementary data from mutagenesis and EPR in our original manuscript. As described above, to

further support the observation from the BiP-ATP2 structure, we have now provided FRET analysis of the polypeptide-binding pocket in the revised manuscript. As described in our revised manuscript, we did observe the presence of a fully-closed L_{1,2} conformation consistent with our BiP-ATP2 structure in solution upon ATP binding. Moreover, our FRET data supported this full-closed L_{1,2} conformation as the major conformation in solution for the ATP-bound state, and further revealed that Hsp40 binding directly shifts Hsp70s from this fully-closed conformation to the open conformation. Please see the sections “**The unique conformation of L_{1,2} observed in the BiP-ATP2 structure is present as a dominant form in solution**” and “**Hsp40 binding shifts the dominant fully-closed conformation of Hsp70 to open conformation**” in the revised manuscript). We believe that we have provided sufficient solution studies to support the presence of this novel conformation in solution and revealed essential mechanism of allostery.

3. *The switch to DnaK for some of the experiments is poorly justified. Was this done because BiP is not amenable to biochemical characterization?*

Response and Change: We apologize for not making the switch from BiP to DnaK well justified. As we described in our original manuscript, the reviewer was correct that the biochemical characterization and chaperone activity assays are better established for DnaK than for BiP. Thus, it is experimentally easier to test the structural hypothesis using DnaK. We have revised our manuscript to better justify the switch from DnaK to BiP. Please see the revised manuscript in both the FRET and mutagenesis sections (**The unique conformation of L_{1,2} observed in the BiP-ATP2 structure is present as a dominant form in solution** and **Conformational impact of three conserved glycine residues in β 1-L_{1,2}- β 2 region**).

4. *The EPR experiment required two mutations: changing the native Cys to Ala, and introducing the new Cys. I didn't see any characterization of the function of these proteins.*

Response and Change: We thank the reviewer for pointing this out. In our original manuscript, we have determined that the substrate binding activity was not affected by either kind of mutations. For the C15A modification in DnaK, we have published a detailed characterization this year and referenced it in our revised manuscript (Liu et al., Cell Stress Chaperones (2017) 22(2):201-212). Briefly, this C15A modification has little biochemical effect on DnaK and the chaperone activity is largely unaffected. As the reviewer suggested, we have carried out more functional characterization on the rest of the mutations including refolding assay for *in vitro* chaperone activity for DnaK, and determining both intrinsic ATPase activity and peptide stimulation on ATPase activity for BiP. Please see the revised manuscript (Supplementary Fig. 10c, d, h, i). In addition, for the FRET analysis with DnaK protein, we have included two modifications: M404C and T437C, besides the aforementioned C15A modification. We have characterized these two modifications to show that neither chaperone activity nor biochemical activity was affected appreciably (please see Supplementary Fig. 7 in the revised manuscript).

5. *Thus, this paper is not acceptable as is, and could be improved. But if the data are not used to propose a mechanism of allostery, which seems a stretch, then it is not clear how broadly read it will be. It may not be appropriate for Nature Communications, even if revised.*

Response and Change:

First, in response to this comment, we have made two major changes to our manuscript: adding two new sections in the RESULTS with two new figures.

a) Besides the mutagenesis and EPR studies, we now have provided further solution studies using FRET to support the presence of the new conformation observed in our BiP-ATP2 structure in solution as a major form for the ATP-bound state (please see the section “**The unique conformation of L_{1,2} observed in the BiP-ATP2 structure is present as a dominant form in solution**” and Fig. 3 in the revised manuscript).

b) We have extended our FRET analysis to test the effect of Hsp40 co-chaperone on the conformation landscape of Hsp70s. Our results suggested that Hsp40 binding directly shifts the polypeptide-binding pocket from mainly fully-closed conformation to mostly open conformation (Please see the section “**Hsp40 binding shifts the dominant fully-closed conformation of Hsp70 to open conformation**” and Fig. 6 in the revised manuscript).

Secondly, based on the suggestions from both reviewers, we have substantially revised our manuscript to focus on mechanism of allostery, and stress the mechanistic insights and importance of our finding to chaperone activity and chaperone cycle.

Therefore, together with the new BiP-ATP2 structure, we believe that our FRET, mutagenesis, and EPR analysis have provided convincing evidence for the novel and paradigm-shifting mechanism of Hsp70 allostery and chaperone cycle revealed by this study:

1) The fully-closed conformation of L_{1,2} observed in our BiP-ATP2 structure is present as the dominant conformation in solution for Hsp70s in the ATP-bound state.

Although the BiP-ATP2 structure carries a L_{3,4} modification, it has provided an invaluable starting point to reveal this novel conformation especially when WT Hsp70s failed to crystallize. This conformation is different from all previously published Hsp70-ATP structures which strictly depend on the ATPase-deficient mutation T199A. T199A mutation may shift the conformational equilibrium toward the open conformation.

2) This new conformation is unique for the ATP-bound state for Hsp70s, and is essential for the chaperone cycle and chaperone activity of Hsp70s. Thus, the polypeptide-binding pocket is more dynamic in the ATP-bound state than that of the ADP-bound and nucleotide-free states.

3) This fully-closed conformation of L_{1,2} is incompatible with binding polypeptide substrate. As the dominant conformation in solution for the ATP-bound state, this conformation supports an active release of bound substrate upon ATP binding to Hsp70s.

4) Binding of Hsp40 co-chaperone directly shifts this fully-closed L_{1,2} conformation to the open conformation while Hsp70s remain in the ATP-bound state. Thus, Hsp40 is the key to promote efficient substrate binding in the ATP-bound state to initiate productive chaperone cycle.

5) Taken together, we proposed a refined chaperone cycle with three novel and paradigm-shifting features (Please see Discussion in the revised manuscript for details):

a) In the ATP-bound state, the polypeptide-binding site is dynamic, and can assume at least three different conformations: full-closed, open, and partially open. The fully-closed conformation is the dominant conformation.

b) Hsp40 co-chaperone is the key signal to initiate efficient substrate binding and thus productive chaperone cycle in Hsp70s by shifting the fully-closed conformation to either the open or partially open conformations.

c) Upon ATP rebinding following ATP hydrolysis, this new ATP-bound state is mainly in the fully-closed conformation, propelling bound substrate to release from Hsp70s. This active release provides an opportunity for the released substrate to fold and new substrate to bind.

Taken together, we have proposed: 1) a novel, unexpected, and active mechanism for polypeptide substrate release caused by ATP-induced allostery; 2) a novel mechanism for productive substrate binding and chaperone cycle: Hsp40 is the key in regulating the allostery in Hsp70s.

In summary, we believe that our findings in the revised manuscript have directly addressed the molecular mechanism of the ATP-induced allostery in Hsp70s, and provided paradigm-shifting insights in the mechanism of Hsp70 chaperone activity.

B. Reviewer 2:

This reviewer thinks that “Overall, the manuscript is of good quality and the new BiP conformation obtained by the authors seems to underline the structural flexibility of Hsp70s and adds to the understanding of Hsp70 chaperones.” However, “there are a number of points that need to be addressed by the authors”. Below, we address the reviewer’s concerns point-by-point.

1. The L_{3,4} modification prevents dimerization of BiP and affects substrate binding of Hsp70s (see e.g. Rist et al. 2006). Therefore, this variant is a matter of controversy in the field as to what extent it can be considered as an intact Hsp70. The authors should discuss this in the manuscript. How do the authors exclude that the novel conformation is due to using this variant?

Response and Change: We agree with the reviewer that the L_{3,4} modification affects substrate binding. So far, there is no data suggesting that this modification affects dimerization in the ATP-bound state although this mutation prevents the non-functional oligomerization of Hsp70s in the ADP-bound and apo states.

We appreciate the concern from the reviewer on “to what extent this mutant can be considered as an intact Hsp70”. That was also our concern when we first used this mutant.

1) We agree with both reviewers that only structure is not enough. As described above in our responses to the first concern from Reviewer 1, to test whether this novel conformation observed in our new BiP-ATP2 structure exists in solution and is not due to the L_{3,4} modification, we have carried out mutagenesis and EPR analysis in our original manuscript. Now we have provided FRET support in the revised manuscript. Please see the newly added section “**The unique conformation of L_{1,2} observed in the BiP-ATP2 structure is present as a dominant form in solution**” in the RESULTS. We believe that these complementary studies have provided strong support for the presence of this novel conformation in normal Hsp70s in solution. Thus, this new structure has provided an invaluable starting point and novel hypothesis for us to test using solution assays although it carries the L_{3,4} modification.

2) This mutant was designed based on the Sse1, a distant relative of Hsp70. Ideally, we would like to obtain Hsp70 structures without any modification. However, for classic Hsp70s, this is challenging and has not been possible. So far, three classic Hsp70 structures in the ATP-bound state have been published: two for DnaK, and one for BiP. Recently, a structure of Ssb, a specialized Hsp70 in fungi, in complex with ATP has been published. All these structures strictly depend on at least two modifications present at the same time: 1) the ATPase-deficient mutation T199A in DnaK (T229A in BiP); and 2) either an engineered disulfide bond designed based on the Sse1 structure or the L_{3,4} mutant. Compared to all these structures, our new BiP-ATP2 structure does not have this ATPase-deficient mutation T199A in the NBD. In another word, our new BiP-ATP2 structure has one less important modification.

Except for our new BiP-ATP2 structure reported in this manuscript, the T199A mutation was critical for all the published structures of Hsp70 in the presence of ATP. T199A not only completely abolishes chaperone activity but also totally eliminates Hsp40 interaction. How this mutation represents the ATP-bound state is not fully understood.

3) In addition, as the reviewer is aware, we have shown previously that the L_{3,4} modification does not affect either the structural integrity of the SBD or the polypeptide-binding pocket by solving structures of the isolated domains with and without this modification (Yang et al., Structure (2015) 23:2191-2203). In the isolated SBD structure carrying this modification, the model peptide NR binds to the polypeptide-binding pocket in a virtually identical way as that of the WT, suggesting the L_{3,4} modification has no appreciable effect on the polypeptide-binding pocket besides the deletion of L_{3,4} although its affinity for substrate is significantly reduced. We have included this information in Supplementary Fig. 3 in the revised manuscript.

2. Along these lines, the BiP L3,4 variant should be included in the fluorescence anisotropy assay (Fig. 4b) to provide a better correlation of the structure with the biochemical data.

Response and Change: We thank the reviewer for pointing this out. As the reviewer suggested, we have revised our manuscript to include these data (please see Supplementary Fig. 3 in the revised manuscript). Since the result of WT control is slightly different from that in Fig. 4b, we have included these data in the Supplementary Fig. 3.

3. A complete structure based sequence alignment should be added to the supplement.

Response and Change: As the reviewer suggested, we have added a complete structure based sequence alignment to the supplemental material. Please see the revised manuscript (Supplementary Fig. 1).

4. The X-ray structure seems overall of good quality – based on the data collection and refinement statistics. However, when reading the validation report, there are a few issues that the authors have to explain:

1) The structure was refined using Phenix and Refmac, but the geometries have not been properly restrained as the bond length variation and the angles are by far too high. Please explain. Along this line - the authors should provide the Molprobit output.

2) The I/ σ I is 2.6, which indicates that the crystals diffract to higher resolution than the cutoff used by the authors. The data should be cut according to the cc1/2 criteria (Karplus and Diederichs, Science 2012). The authors should provide a cc1/2 value to explain this discrepancy. The redundancy of the structure is low,

probably due to the low symmetry (P1). Merging different crystals or using separately collected high and low resolution datasets might be an option to improve the data - if available.

Response and Change:

1) We sincerely apologize for this oversight and thank the reviewer for pointing it out. We have carried out new refinement with more properly restrained geometries on a dataset with higher redundancy (see below). The r.m.s. deviations of bond lengths and angles are good. Please see the revised Table 1 and validation report. As the reviewer suggested, we have run Molprobit, and provided the output (please see the attached Molprobit reports).

2) We thank the reviewer for this constructive suggestion. As the reviewer suggested, we have provided the $cc1/2$ value, which is good for the dataset (please see Table 1). The reason that we cut $I/\sigma I$ at 2.6 is due to the low completeness in the high resolution shell. For resolution shell 1.83-1.86 Å, the completeness is 64.7%, and redundancy is 2.0 although the $cc1/2$ is 0.850.

The reviewer is correct. The low redundancy of the structure is due to the low symmetry. As the reviewer suggested, to increase redundancy, we have combined another dataset collected on the same crystal with the dataset used in our original structure determination and refinement. The redundancy has improved (please see Table 1). The current model was refined with this combined dataset.

5.

1) The methods section is rather superficial, lacking important details. This includes buffers used in the purification, the yeast strains used in the growth assays including Ssa (or at least proper citations). Likewise, the citations for crystallographic software are completely absent (COOT, Phenix, ARP/wArp,...). This is a bad habit as the funding of the developers depends on the citations and the citations should be added.

2) It is not clear how the figures were generated (Pymol??).

3) Figure 2 d-f, Which kt was used to calculate the surfaces of the SDBbeta? Please also cite APBS if used to calculate the surfaces.

Change: We apologize for these oversights.

1) As the reviewer suggested, we have tried our best to add sufficient details to the methods section, and add proper citations to the revised manuscript including the crystallographic software used.

2) The figures were generated using Pymol. We have added this information in the revised manuscript.

3) For these figures, we used Pymol. We have added this information in the revised manuscript.

Minor points:

1. The crystal packing shown in suppl. Fig. 1f (mentioned in the main text line 104) should be extended to clarify whether SDBbeta is involved in crystal contacts.

Change: We thank the reviewer for this constructive suggestion. We have added the suggested crystal contacts in Supplementary Fig. 5 and description in the section “**A surprising new conformation of the polypeptide-binding pocket**” in the revised manuscript (second to the last paragraph). As the reviewer is fully aware, crystal contacts are unavoidable for crystal structures. In fact, all the Hsp70-ATP structures have crystal contacts around the peptide-binding loops. In the two previously published DnaK-ATP structures, there are a number of crystal contacts around SBD β . However, there is only one hydrophobic contact involves L_{1,2} in the BiP-ATP2 structure; in contrast, more extensive crystal contacts involve L_{1,2} in the two DnaK-ATP structures.

2. In the discussion (line 293) a citation should be included as the dynamics of the SBD with respect to ATP binding has been described for DnaK in Rist et al (2006).

Change: As the reviewer suggested, we have added this citation to line 293. Moreover, in our original manuscript, we have referenced this citation in the paragraph above line 293.

3. The figures are labeled in different fonts which are sometimes hard to read (e.g. Fig. 1a,b) and the position of the labels needs improvement (e.g. Fig 5, II, L3,4/L5,6 label in the figure).

Change: We apologize for this oversight and thank the reviewer for pointing this out. We have changed all the fonts to Time New Roman to make all the fonts consistent. We have shifted the positions of a number of labels and hope we have improved the figures.

4. The model provided in Fig. 5 is of poor quality and needs improvement.

Change: We thank the reviewer for this constructive suggestion. We have redrawn the model and hope that the quality of the model has improved. Furthermore, we have added new features based on the newly-added FRET results.

5. Supplementary Fig. 5b: The proper abbreviation for molecular weight should be used (kDa).

Change: We have made the changes and thank the reviewer for pointing it out.

Sincerely yours,

Qinglian Liu, PhD
Associate Professor
Department of Physiology and Biophysics
Virginia Commonwealth University
Richmond, VA 23298

Phone: 804-628-4851

Email: qinglian.liu@vcuhealth.org

Reviewers' comments:

Reviewer #1 (Remarks to the Author):

I applaud the authors' efforts to answer all critiques from the previous review. In particular, their efforts to validate the conformation seen in the BiP-ATP crystal structure as also being populated in solution are laudable.

However, I remain very skeptical that DnaK should be used to interpret BiP behavior. These two are different enough in sequence to raise questions.

I think the paper should be focused on the BiP story, cleaned up for grammar and wording mistakes, and made more accessible by inclusion of only the most relevant data. The authors have made the case that their earlier model for the mode of substrate release based on the crystal structure is compelling. Now simply put in the data that are needed for that story.

Ironically, in an effort to address every point raised in the previous review, they have diluted the impact of the original story.

The argument about the interaction with a J protein as an essential step for substrate binding is very interesting. Can they make this case using data for BiP??

We would like to thank the first reviewer for reviewing our revised manuscript, and his/her additional constructive and insightful comments to improve our manuscript. We are very pleased to learn that we have provided compelling support for our “earlier model for the mode of substrate release based on the crystal structure”, and we greatly appreciate the reviewer’s encouraging comments and compliment on our efforts.

Besides grammar and wording mistakes, the major remaining comment is that our manuscript “should be focused on the BiP story”. We have made every effort to address this important concern. Accordingly, we have made the following changes to our manuscript. As suggested by the editor, we have highlighted all the major changes in the manuscript text file (in yellow).

Comment 1:

However, I remain very skeptical that DnaK should be used to interpret BiP behavior. These two are different enough in sequence to raise questions.

I think the paper should be focused on the BiP story, cleaned up for grammar and wording mistakes, and made more accessible by inclusion of only the most relevant data. The authors have made the case that their earlier model for the mode of substrate release based on the crystal structure is compelling. Now simply put in the data that are needed for that story.

Ironically, in an effort to address every point raised in the previous review, they have diluted the impact of the original story.

Response and Change:

We thank the reviewer for this constructive suggestion. Based on this suggestion, we have made the following changes to our manuscript to focus on BiP and include only the most relevant data:

- 1) To focus on BiP, we have carried out FRET analysis using BiP and showed that the fully-closed conformation of $L_{1,2}$ observed in our BiP-ATP2 structure is the major form in solution for BiP. We have replaced the originally DnaK FRET data in Figure 3 with the new BiP FRET data (please see the revised Figure 3), and revised the section “**The unique conformation of $L_{1,2}$ observed in the BiP-ATP2 structure is present as a dominant form in solution**” in the RESULTS accordingly to describe only the BiP FRET results.
- 2) We have removed the following DnaK data: a) peptide-binding and growth test in the original Figure 4, and b) EPR analysis in the original Figure 5. We have changed the order of Figure 4 and 5 and their corresponding sections in the RESULTS for a better flow of the manuscript. Thus, the revised Figure 4 and the corresponding section “**ATP-binding specifically increases the conformational dynamics of $L_{1,2}$** ” in the RESULTS are focused on the EPR data of BiP to show the dynamics of $L_{1,2}$ in BiP; and the revised Figure 5 and the corresponding section “**Conformational impact of three conserved glycine residues in $\beta 1$ - $L_{1,2}$ - $\beta 2$ region**” in the RESULTS are focused on the influence of G425P and G430P/G431P mutations on BiP’s peptide-binding activity. Since it is difficult to assay BiP’s chaperone activity, we used DnaK for the chaperone activity assay in the revised Figure 5, and this is the only DnaK data for the first six figures. At the same time, we have revised the Supplementary Figures accordingly.

3) We agree with the reviewer that BiP and DnaK are significantly different in sequence and function. We included the FRET analysis of DnaK and the effect of Hsp40 DnaJ on DnaK in the revised Figure 7 to show the functional difference and conservation between BiP and DnaK. Accordingly, we have added a new section in the RESULTS to describe these studies. Please see the last section of the RESULTS (**The conformational landscape of L_{1,2} is largely conserved in *E.coli* Hsp70 DnaK**).

In summary, for the RESULTS, except for the first two sections on the description of the BiP-ATP2 structure, we have made major revisions to the rest of five sections including corresponding figures to focus on BiP.

Comment 2:

The argument about the interaction with a J protein as an essential step for substrate binding is very interesting. Can they make this case using data for BiP??

Response and Change:

We thank the reviewer for this constructive and insightful suggestion. As the reviewer suggested, we have carried out FRET analysis on BiP using ERdj3, a J-protein co-chaperone for BiP. Our results have shown that the interaction with J-proteins as an essential step for substrate binding is conserved for BiP. We have replaced the original DnaK-DnaJ results with these new BiP-ERdj3 studies. Thus, the revised Figure 6 and section “**Binding Hsp40 co-chaperone shifts the dominant fully-closed conformation of Hsp70 to open conformation**” in the RESULTS are focused on these new BiP-ERdj3 results. Please see the revised manuscript.

Comment 3:

cleaned up for grammar and wording mistakes

Response and Change:

We apologize for these oversights. We have tried our best to correct any grammar and wording mistake that we could find.

We hope that we have satisfactorily addressed all the remaining issues raised by the reviewer, adequately improved our manuscript, and made our manuscript suitable for publication on *Nature Communications*.

Sincerely yours,

Qinglian Liu, PhD
Associate Professor
Department of Physiology and Biophysics
Virginia Commonwealth University
Richmond, VA 23298
Phone: 804-628-4851
Email: qinglian.liu@vcuhealth.org

Reviewers' Comments:

Reviewer #1 (Remarks to the Author):

The authors have constructively addressed my criticisms and suggestions.

There remain odd capitalization, a few typos and grammar errors, and inconsistent reference format, but these can be addressed in copyediting.